# Visual-Word Tokenizer: Beyond Fixed Sets of Tokens in Vision Transformers

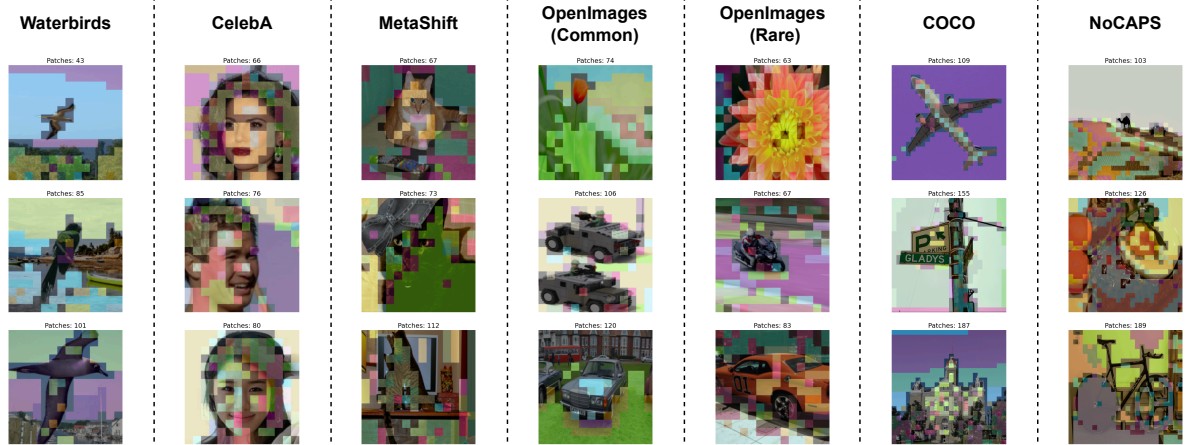

Figure 1: Visualization of patch matching by our visual word tokenizer ($\mathcal{T}_{inter}^{100}$ model). Patches that are matched with one another are indicated by identical colors. Higher patch matching is exhibited by the background rather than foreground across datasets. Patch matching serves as a rudimentary form of image segmentation by grouping similar non-adjacent visual concepts.

## Abstract

The cost of deploying vision transformers increasingly represents a barrier to wider industrial adoption. Existing compression techniques require additional end-to-end fine-tuning or incur a significant drawback to runtime, making them ill-suited for online (real-time) inference, where a prediction is made on any new input as it comes in. We introduce the **Visual Word Tokenizer** (VWT), a training-free method for reducing energy costs while retaining performance and runtime. The VWT groups visual subwords (image patches) that are frequently used into visual words while infrequent ones remain intact. To do so, *intra*-image or *inter*-image statistics are leveraged to identify similar visual concepts for sequence compression. Experimentally, we demonstrate a reduction in wattage of up to 25% with only a 20% increase in runtime at most. Comparative approaches of 8-bit quantization and token merging achieve a lower or similar energy efficiency but exact a higher toll on runtime (up to 100% or more). Our results indicate that VWTs are well-suited for efficient online inference with a marginal compromise on performance.

## 1 Introduction

In recent years, deep learning has seen continuous integration into a variety of systems worldwide. From coding to gaming, neural networks are increasingly deployed in online scenarios where asynchronous requests are processed in real-time. However, due to the size and complexity of modern architectures, such models are costly to run in practice. To address this, various methods have been proposed to improve model efficiency such as Knowledge Distillation (Hinton et al., 2015), Pruning (Han et al., 2015; Michel et al., 2019), and

Quantization (Dettmers et al., 2022). However, many of these methods either require end-to-end fine-tuning to recover performance or significantly reduce runtime. In the field of Natural Language Processing (NLP), there is a growing trend towards improving efficiency via tokenization (Gee et al., 2022; 2023a; Dagan et al., 2024; Yamaguchi et al., 2024; Minixhofer et al., 2024). Newer large language models (LLMs) (Team et al., 2024; Dubey et al., 2024) exhibit a noticeably larger vocabulary than their earlier counterparts (Devlin, 2018; Radford et al., 2019), thereby producing shorter sequences across various distributions. For computer vision, increasing interest is placed on reducing the cost of deploying the vision transformer (ViT) (Dosovitskiy et al., 2020). As image encoders in larger vision-language systems, ViTs are used to process images as fixed sets of tokens. Similar to downsampling in convolutional neural networks, most research (Kong et al., 2022; Liang et al., 2022; Rao et al., 2021; Marin et al., 2021; Bolya et al., 2022; Bian et al., 2023; Kim et al., 2024) has focused on merging and/or pruning tokens in the intermediate layers to reduce computational overhead. Given the analogous architecture of the transformer across image and text modalities, our work looks instead at the idea of tokenization for efficiency by splitting an image into variable sets of tokens.

To introduce variability, we draw inspiration from subword tokenization algorithms (Gage, 1994; Sennrich et al., 2016) used in NLP which follow the principle that common words should remain intact while infrequent ones are broken down into meaningful subword units. Instead of a top-down approach – splitting words into subwords, our work for image data takes a bottom-up approach by grouping visual subwords (image patches) into visual words. We also twist the underlying principle: frequently used patches should be grouped as they are more likely to describe common features while infrequent ones remain intact as they might carry task-relevant information. We propose two procedures to capture this principle. The first is an *intra*-image approach where patches with the lowest pixel variance within each image are grouped as they typically represent uniform areas (e.g. backgrounds). The second is an *inter*-image approach where basic features across multiple images such as colors or edges are discovered as visual words. Image patches are then grouped based on the similarity of these basic characteristics. Crucially, patches that have distinct characteristics (i.e. high dissimilarity with any visual word) remain intact and form separate visual subwords.

## 2 Related Work

**Efficient ViTs.** Most works for improving the efficiency of ViTs have focused on reducing tokens in the intermediate layers by leveraging importance scores. In Liang et al. (2022); Xu et al. (2022); Kong et al. (2022); Bian et al. (2023), redundancy is addressed by fusing tokens. Both Rao et al. (2021) and Tang et al. (2022) opt to prune such tokens instead. Recent efforts (Cao et al., 2023; Chen et al., 2023; Bonnaerens & Dambre, 2023; Kim et al., 2024) attempt to combine the benefits of merging and pruning. In Tran et al. (2024), an additional metric termed the energy score is used to better identify redundancy. Uniquely, Fayyaz et al. (2022) use inverse transform sampling to select important tokens. Most relevant to our work are Marin et al. (2021) and Bolya et al. (2022). The former assigns tokens to centroids via clustering, while the latter progressively merges tokens layer-by-layer in a training-free manner[1].

**Specialized Tokenizers.** Our method also takes inspiration from efficient inference in NLP. Increasingly, the tokenizer's vocabulary is specialized to reduce the input token length. In Gee et al. (2022), domain adaptation of the tokenizer ensures fewer subword or character tokens are produced. Gee et al. (2023a) followed up by introducing n-grams for tokenization beyond the word-level boundary. In Dagan et al. (2024), tokenizer specialization is also shown to accelerate the task of code generation with modern LLMs. Meanwhile, Yamaguchi et al. (2024) analyzed the effectiveness of various vocabulary adaptation techniques for efficient cross-lingual inference. Recently, Minixhofer et al. (2024) leveraged hypernetworks for zero-shot tokenizer transfer of newly domain-adapted vocabularies.

**Vector Quantization.** The idea of discretizing continuous distributions has been explored in many works, most recently for image generation. Yang et al. (2007) leveraged clustering for learning a codebook that maps keypoint descriptors to discrete visual words. In Wu et al. (2020) and Bao et al. (2021), discretization is applied as part of the modelling for ViTs. Van Den Oord et al. (2017) learned discrete image representations by introducing the *Vector Quantised-Variational Autoencoder* (VQ-VAE) approach. Esser et al. (2021) and

---

[1]Unlike Bolya et al. (2022), we do not include Marin et al. (2021) as one of our baselines due to a lack of code release.

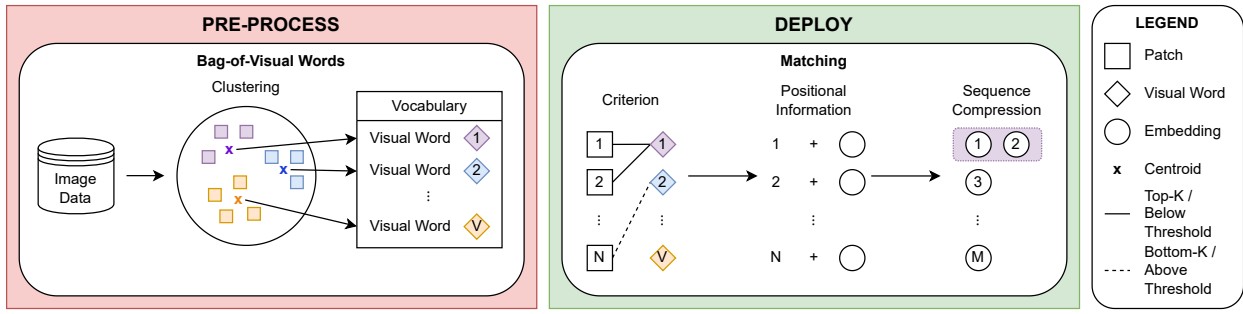

Figure 2: Overview of the Visual Word Tokenizer. An *intra*-image approach. In the forward pass, the pixel variance of the patches is computed with the top-k lowest values being masked (DEPLOY). The grouped tokens are dropped after positional information is added. Note that $\mathcal{V} = 1$ as only a single pixel variance feature is used. An *inter*-image approach. A Bag-of-Visual Words is formed by clustering patches in the pixel space (PRE-PROCESS). In the forward pass, the minimum pairwise cosine distance between the patches and visual words is computed with values above the threshold being masked (DEPLOY). The grouped tokens are averaged after positional information is added.

Yu et al. (2021) further improved upon the VQ-VAE by combining the expressiveness of transformers with an adversarial framework. Increasingly, vision-language models paired with codebooks conduct image synthesis autoregressively (Ramesh et al., 2021; Yu et al., 2022; Lu et al., 2022; 2024; Team et al., 2023; Team, 2024; Sun et al., 2024). Lastly, Yang et al. (2022) tackled disentangled representation learning and scene decomposition by tokenizing images into separate visual concepts.

## 3    Visual Word Tokenizer

In ViTs, tokenization is a process that splits an image into patches (tokens) which are then projected into a series of embeddings. The number of patches is typically fixed (e.g. 197) based on the choice of architecture. We seek to split an image into variable length inputs instead (i.e. certain images will use 80 tokens, while others a 100). This variability is induced by conventional text tokenizers (Gage, 1994; Sennrich et al., 2016; Wu et al., 2016; Kudo & Richardson, 2018) for model efficiency. We propose to achieve this via visual words that group patches (visual subwords) based on some commonality in the pixel space. These visual words capture frequently used patches while retaining infrequent ones as is. A simple yet effective grouping can be done using either a criterion that looks at statistics of only one image (an *intra*-image approach) or across many images (an *inter*-image approach). Figure 2 summarizes the **Visual Word Tokenizer** (VWT).

### 3.1    An *intra*-image approach

The pixel variance of the image patches is the simplest criterion that can be used for grouping. In Figure 2, this approach only utilizes the DEPLOY step by grouping the top-k patches with the lowest pixel variance while leaving the rest intact. To compress the grouped tokens, we opt to drop them as they tend to exhibit excessive homogeneity. We do not include the [CLS] token for dropping. This approach is inspired by Minderer et al. (2024) which aimed to reduce the training cost of OWLv2, an open-vocabulary object detector. Dropping patches with the lowest pixel variance removes padding areas and uniform backgrounds from the input mosaics of raw images, thereby increasing training efficiency.

### 3.2    An *inter*-image approach

Inspired by text tokenizers and codebooks, we propose a variable-length approach that statistically discovers visual words across many images. The tokenizer consists of two steps: PRE-PROCESS and DEPLOY

**Pre-Process.**    The Bag-of-Visual Words (BoVW) is a popular method for modeling images via discrete representations. In Yang et al. (2007), k-means clustering is applied to keypoint descriptors from SIFT (Lowe,

| Dataset | Model | $Base$ | $\mathcal{T}_{intra}^{0.5}$ | $\mathcal{T}_{inter}^{100}$ | | $\mathcal{T}_{inter}^{1000}$ | | $\mathcal{T}_{inter}^{10000}$ | |
|---|---|---|---|---|---|---|---|---|---|
| | | | | In-Domain | ImageNet | In-Domain | ImageNet | In-Domain | ImageNet |
| Waterbirds | CLIP | 197 | 99 | 125 | 124 | 144 | 144 | 165 | 169 |
| CelebA | | | | 89 | 88 | 130 | 119 | 163 | 155 |
| MetaShift | | | | 112 | 109 | 136 | 135 | 162 | 164 |
| OpenImages (Com.) | | | | - | 114 | - | 136 | - | 162 |
| OpenImages (Rare) | | | | - | 110 | - | 133 | - | 160 |
| COCO | BLIP | 577 | 289 | 267 | 264 | 317 | 312 | 408 | 405 |
| NoCaps | | | | - | 257 | - | 307 | - | 403 |

Table 1: Token length per sample (including [CLS]). $\mathcal{T}_{inter}^{\mathcal{V}}$ of varying pre-processing data and vocabulary sizes are shown. Unlike text tokenizers, domain specialization (i.e. In-Domain) does not result in greater compression. Smaller vocabularies produce shorter sequences whereas larger vocabularies result in patches being increasingly matched to separate visual words.

| Dataset | Model | $Base$ | $Q_8$ | $ToME$ | $\mathcal{T}_{intra}^{0.5}$ | $\mathcal{T}_{inter}^{100}$ | $\mathcal{T}_{inter}^{1000}$ | $\mathcal{T}_{inter}^{10000}$ |
|---|---|---|---|---|---|---|---|---|
| Waterbirds | CLIP | 123.99 | 73.89 | 91.64 | 110.93 | 102.65 | 107.93 | 117.36 |
| CelebA | | 126.53 | 74.47 | 93.07 | 114.37 | 96.31 | 102.12 | 115.46 |
| MetaShift | | 123.30 | 74.02 | 90.96 | 113.84 | 98.98 | 105.78 | 114.70 |
| OpenImages (Com.) | | 135.05 | 74.80 | 94.63 | 114.62 | 106.33 | 112.67 | 123.95 |
| OpenImages (Rare) | | 135.66 | 75.27 | 94.04 | 115.78 | 104.73 | 110.75 | 123.05 |
| COCO | BLIP | 191.74 | 81.68 | 147.23 | 169.77 | 144.71 | 156.25 | 171.84 |
| NoCaps | | 185.12 | 81.89 | 149.26 | 175.06 | 141.20 | 155.11 | 170.38 |

(a) Wattage (watt) ↓

| Dataset | Model | $Base$ | $Q_8$ | $ToME$ | $\mathcal{T}_{intra}^{0.5}$ | $\mathcal{T}_{inter}^{100}$ | $\mathcal{T}_{inter}^{1000}$ | $\mathcal{T}_{inter}^{10000}$ |
|---|---|---|---|---|---|---|---|---|
| Waterbirds | CLIP | 8.48 | 88.55 | 17.04 | 5.84 | 9.88 | 9.73 | 9.91 |
| CelebA | | 8.33 | 89.41 | 17.25 | 5.86 | 9.45 | 9.54 | 9.99 |
| MetaShift | | 8.21 | 89.48 | 17.23 | 6.01 | 9.99 | 9.87 | 9.75 |
| OpenImages (Com.) | | 8.83 | 92.23 | 17.44 | 5.97 | 9.64 | 9.69 | 10.23 |
| OpenImages (Rare) | | 8.60 | 86.00 | 17.04 | 6.19 | 9.69 | 10.02 | 9.67 |
| COCO | BLIP | 12.45 | 57.11 | 15.06 | 8.10 | 8.73 | 9.10 | 10.31 |
| NoCaps | | 12.44 | 57.61 | 15.08 | 8.04 | 9.49 | 9.28 | 10.51 |

(b) Runtime (millisecond) ↓

Table 2: Wattage and runtime per sample. Compared to *Base*, VWTs reduce wattages by up to 25% with $\mathcal{T}_{inter}^{100}$. Although $Q_8$ and $ToME$ display lower wattages than VWTs, they induce significantly longer runtimes, especially the former. $\mathcal{T}_{intra}^{0.5}$ is consistently faster than *Base*, while $\mathcal{T}_{inter}^{10000}$ may increase runtimes by up to 20% at most (except on COCO and NoCaps).

2004) to learn a fixed set of centroids. These centroids represent the vocabulary to which multiple descriptors are mapped in a process termed *Vector Quantization* (VQ). In our method, we adopt a variation of this framework by building the BoVW using patches within the pixel space. Our design choice is motivated by two main factors. First, we find keypoint descriptors to be costly for inference. In each forward pass, computing keypoints for each image would significantly increase runtime. Second, in our early experimentation, we observed that patches in the embedding space have little similarity to one another. Such issues were also described by Bolya et al. (2022), thus leading to their use of attention scores instead. Further justification for leveraging the pixel space is provided in Section 4.6.

| Model | Waterbirds | | CelebA | | MetaShift | | OpenImages | |
|---|---|---|---|---|---|---|---|---|
| | Average ↑ | Worst ↑ | Average ↑ | Worst ↑ | Average ↑ | Worst ↑ | Common ↑ | Rare ↑ |
| $Base$ | 79.06 | 21.86 | 89.61 | 47.21 | 95.31 | 87.69 | 70.48 | 63.36 |
| $Q_8$ | 79.94 | 24.25 | 89.73 | 48.19 | 95.23 | 88.21 | 70.48 | 63.19 |
| $ToME$ | 79.80 | 27.05 | 89.25 | 28.52 | 94.24 | 87.37 | 65.69 | 60.05 |
| $\mathcal{T}_{intra}^{0.5}$ | 75.56 | 31.00 | 89.27 | 50.93 | 92.94 | 86.15 | 65.52 | 59.73 |
| $\mathcal{T}_{inter}^{100}$ | 78.41 | 26.01 | 90.04 | 53.15 | 90.24 | 84.28 | 62.90 | 58.10 |
| $\mathcal{T}_{inter}^{1000}$ | 79.19 | 23.83 | 90.79 | 47.96 | 93.94 | 86.67 | 66.15 | 60.56 |
| $\mathcal{T}_{inter}^{10000}$ | 79.68 | 22.90 | 90.12 | 46.85 | 94.81 | 86.15 | 69.03 | 62.50 |

(a) CLIP (image classification and subgroup robustness)

| Model | COCO | | NoCaps | | | | | | | |
|---|---|---|---|---|---|---|---|---|---|---|
| | Karpathy ↑ | | In-Domain ↑ | | Near-Domain ↑ | | Out-of-Domain ↑ | | Overall ↑ | |
| | BLEU@4 | CIDEr | CIDEr | SPICE | CIDEr | SPICE | CIDEr | SPICE | CIDEr | SPICE |
| $Base$ | 34.00 | 107.35 | 103.57 | 14.60 | 98.87 | 14.11 | 92.96 | 13.30 | 98.34 | 14.02 |
| $Q_8$ | 33.85 | 106.87 | 103.86 | 14.53 | 100.07 | 14.18 | 92.72 | 13.49 | 99.12 | 14.10 |
| $ToME$ | 30.02 | 94.40 | 97.35 | 14.42 | 90.74 | 13.39 | 84.22 | 12.69 | 90.37 | 13.40 |
| $\mathcal{T}_{intra}^{0.5}$ | 32.80 | 104.37 | 99.86 | 14.20 | 94.21 | 13.73 | 89.52 | 13.12 | 94.07 | 13.68 |
| $\mathcal{T}_{inter}^{100}$ | 31.10 | 97.70 | 95.58 | 13.78 | 89.62 | 13.11 | 82.42 | 12.44 | 89.02 | 13.08 |
| $\mathcal{T}_{inter}^{1000}$ | 32.12 | 101.79 | 98.19 | 13.95 | 93.85 | 13.49 | 86.03 | 12.67 | 92.89 | 13.39 |
| $\mathcal{T}_{inter}^{10000}$ | 33.18 | 105.08 | 102.84 | 14.59 | 98.03 | 13.95 | 91.52 | 13.14 | 97.40 | 13.88 |

(b) BLIP (image captioning)

Table 3: Zero-shot (training-free) image classification (CLIP), subgroup robustness (CLIP) and captioning (BLIP). VWTs maintain average accuracy relative to *Base* while improving robustness on Waterbirds. On COCO and NoCaps, VWTs retain a higher performance than *ToME*.

Given a dataset, we first patchify the images using the same patch size as the image encoder (e.g. 16 for ViT-B/16). We then cluster the patches via k-means to form the BoVW of a given vocabulary size, where each centroid represents a visual word. Patchification is done via basic tensor operations and not the pre-trained convolutional layer of the ViT to avoid projection into embeddings. We also execute this process in batches using the MiniBatchKMeans[2] algorithm due to memory constraints. Note that MiniBatchKMeans uses the Euclidean distance by design. Since clustering is done in the pixel space, the BoVW may be reused by other ViTs with the same patch size regardless of model type.

**Deploy.** Once the BoVW is formed, we turn towards the process of sequence compression. One way of leveraging the BoVW would be to merge similar patches in the pixel space before projecting them into embeddings. However, such a naive approach will significantly degrade performance as the initialization of a new linear layer for projection is required. To avoid this, we begin by patchifying and computing the pairwise cosine distance between the patches and BoVW. For each patch, we retain only the minimum distance. Unlike text, we are unable to obtain exact matches with images. As such, distances higher than a given threshold are masked out to ensure dissimilar patches are not merged. At this point, we have determined the groupings of similar patches via their connections to the same visual words. We then apply the pre-trained convolutional layer of the ViT on the original image to patchify and project it into a series of embeddings. Before merging, we ensure that positional information is added to the embeddings as we

---

[2]from sklearn.cluster import MiniBatchKMeans

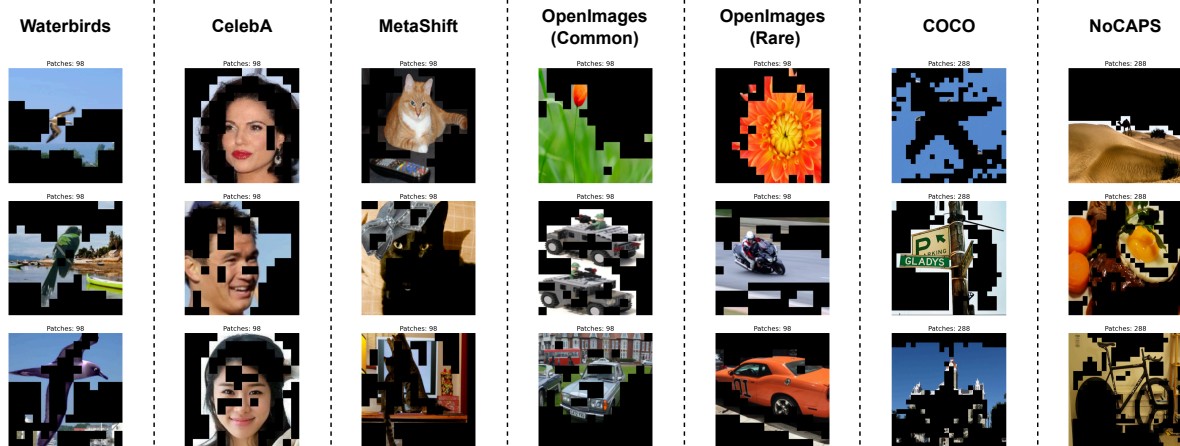

Figure 3: Visualization of patch dropping by our visual word tokenizer ($\mathcal{T}_{intra}^{0.5}$ model). Patches with the lowest pixel variance that are dropped are indicated in black. In most cases, the dropped patches correspond to the background object on average which is typically uninformative.

found it to work better than adding them later. Lastly, we average the embeddings element-wise based on the earlier defined groupings. We do not include the [CLS] token for merging.

For the *inter*-image approach, if batching instead of online inference is desired, the uniform requirement of tensors becomes a challenge. To maintain parallelism, any reduction has to be equal across samples. Due to the non-uniformity of tokenization, sequences have to be sequentially compressed before padding to the same length. We opt to append additional [PAD] tokens until the longest length within the batch is achieved. Similar to text transformers (Vaswani et al., 2017), the attention scores are set to negative infinity before the softmax to nullify the influence of padding. We do not add positional information to the [PAD] tokens as extrapolating such information to non-uniform sequences will significantly worsen model efficiency.

## 4 Experiments

Consider a pre-trained image encoder $f \in \mathcal{F}$ with parameters $\theta \in \Theta$ that transforms inputs $x \in \mathcal{X} \subseteq \mathbb{R}^d$ to encodings $\hat{x} \in \hat{\mathcal{X}} \subseteq \mathbb{R}^{\hat{d}}$. The encodings can then be mapped to labels $y \in \mathcal{Y}$ for some given task, be it classification or generation. More specifically, the ViT first transforms inputs $x$ to tokens $t_1, \ldots, t_N$ before further processing by the attention layers. The number of tokens $N$ is a constant defined by $(\frac{I}{P})^2$, where $I$ and $P$ are the image and patch sizes, respectively. Let $\mathcal{T}$ be the VWT associated with a vocabulary $v \in \mathcal{V}$, where $v$ is a visual word learned from some dataset $D$. The tokenizer transforms the input $x$ into tokens $t_1, \ldots, t_M$, where $M \ll N$. In our experiments, we seek to analyze the effect of $\mathcal{T}$ on online inference efficiency. We focus on the zero-shot setting by eschewing any form of end-to-end fine-tuning for $f$.

### 4.1 Datasets and Settings

We conduct our analysis through the lens of (i) classification performance of visual recognition, (ii) subgroup robustness, and (iii) generative performance of visual captioning. For (i) and (ii), we utilize three publicly available datasets (Waterbirds (Wah et al., 2011), CelebA (Liu et al., 2015), MetaShift (Liang & Zou, 2022)) that are typical benchmarks in robustness and fairness research (Sagawa et al., 2019; Liu et al., 2021; Yang et al., 2023). To perform (zero-shot) classification, we compute the cosine similarity between an image embedding and the following encoded text labels[3] for Waterbirds, CelebA, and MetaShift, respectively: {'landbird', 'waterbird'}, {'non-blond', 'blond'}, {'dog', 'cat'}. Further details on the defined subgroups are provided in Appendix A.1. For (i), we also conduct a large-scale evaluation on the OpenImages v6 dataset (Kuznetsova et al., 2020). Following Huang et al. (2023), the test split is divided into *common* and *rare* subsets that consist of 57 224 and 21 991 images, respectively. The former has 214 classes, while the latter has

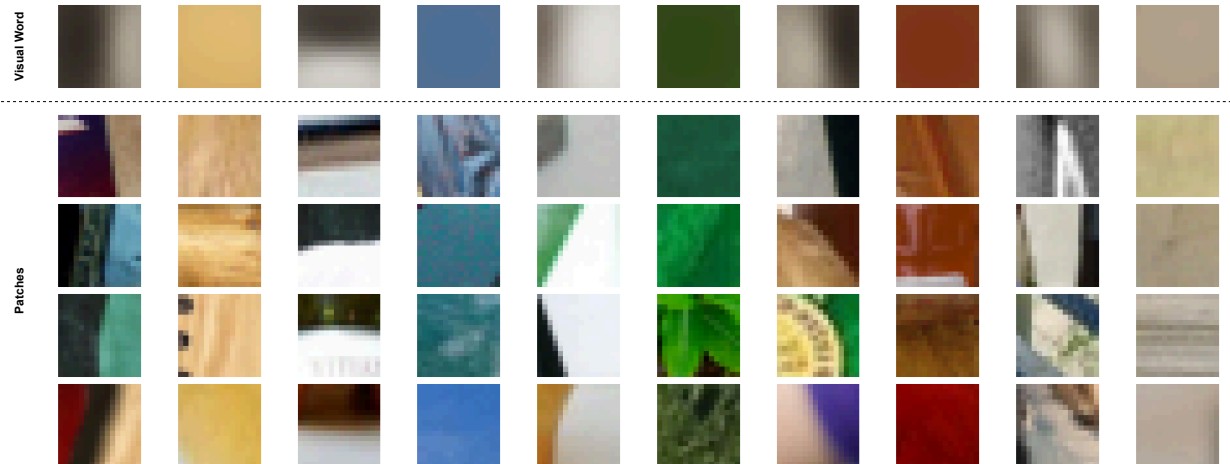

Figure 4: Visualization of the vocabulary of $\mathcal{T}_{inter}^{100}$. Patches from ImageNet-1K are matched to the closest visual word using the Euclidean distance. Visual words are shown to depict basic features such as colors or edges. Each visual word is an average representation of patches that belong to the matched cluster.

200. To perform (zero-shot) classification, we compute the cosine similarity between an image embedding and the encoded text label[3]. For (iii), we utilize the Karpathy test split of COCO dataset (Lin et al., 2014) and a validation set of NoCaps dataset (Agrawal et al., 2019) following the setting in previous work (Li et al., 2022). Finally, to study inference efficiency, we utilize all datasets for the visual tasks (i)-(iii) described above. We compute the efficiency per sample and average across all samples. For the wattage calculation, we call "pynvml.nvmlDeviceGetPowerUsage(handle)/1000" as done by Weights and Biases, which leverages the NVIDIA Management Library for monitoring and managing the NVIDIA GPUs. The experiments are also conducted in an isolated environment, thus ensuring the efficiency measurements are as accurate as possible. Note that we do not include the PRE-PROCESS step of the *inter*-image approach into our efficiency calculations as this process is only done once and may be reused multiple times.

## 4.2 Implementation Details

For image classification, we load the pre-trained CLIP (Radford et al., 2021) model from HuggingFace[4]. An image size of $224 \times 224$ is used with bilinear interpolation for CLIP. For image captioning, we load the pre-trained BLIP (Li et al., 2022) model from HuggingFace[5]. To perform zero-shot captioning, we use a beam size of 3 along with maximum and minimum lengths of 20 and 5, respectively. An image size of $384 \times 384$ is used with bicubic interpolation. Both CLIP and BLIP utilize the ViT-B/16 image encoder unless stated otherwise. Aside from the pre-trained model which we denote as *Base*, we also consider 8-bit quantization (Dettmers et al., 2022) and token merging (Bolya et al., 2022) as additional baselines. We denote the former as $Q_8$ and the latter as *ToME*. Following Bolya et al. (2022), we utilize a reduction per layer for *ToME* of 13 with CLIP and 23 with BLIP due to their respective input image sizes. For the VWTs, we set the top-k of the *intra*-image approach to 50% of the total number of patches which we denote as $\mathcal{T}_{intra}^{0.5}$. For the *inter*-image approach, we set the threshold to 0.1 unless stated otherwise and denote it as $\mathcal{T}_{inter}^{\mathcal{V}}$, where $\mathcal{V}$ is the size of the vocabulary. Lastly, our experiments are conducted using a single NVIDIA A100 GPU. Since our focus is on the **online setting** (real-time), we set the batch size to 1 unless stated otherwise.

## 4.3 VWTs and Inference Efficiency

First we analyze the effects of VWTs on token length. We seek to understand how the choice of pre-processing data and vocabulary size affects the degree of compression. Table 1 shows the token length per sample

---

[3]The prefix "a photo of a " is also added to encode each text label.

[4]https://huggingface.co/openai/clip-vit-base-patch16

[5]https://huggingface.co/Salesforce/blip-image-captioning-base

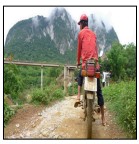 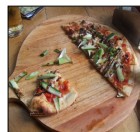 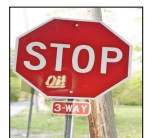 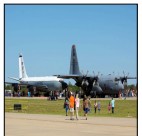

**Base:** a man riding a motorcycle down a dirt road

**Q$_8$:** a man riding a motorcycle down a dirt road with a mountain in the background

**ToME:** a man riding a motorcycle down a dirt road with a mountain in the background

**T$_{intra}$$^{0.5}$:** a man riding a motorcycle down a dirt road

**T$_{inter}$$^{100}$:** a man riding a motorcycle down a dirt road with a mountain in the background

**T$_{inter}$$^{1000}$:** a man riding a motorcycle down a dirt road with a mountain in the background

**T$_{inter}$$^{10000}$:** a man riding a motorcycle down a dirt road with a mountain in the background

---

**Base:** two slices of pizza sitting on a wooden cutting board on a table with a glass of beer in the pizza is on a wooden cutting board

**Q$_8$:** two slices of pizza sitting on a wooden cutting board on a wooden table

**ToME:** two slices of pizza sitting on a wooden cutting board on a wooden table with a glass of beer in the pizza is on a wooden cutting board

**T$_{intra}$$^{0.5}$:** a pizza on a wooden cutting board with a slice taken out of it and a glass of beer on the side of the plate in the background

**T$_{inter}$$^{100}$:** the pizza is on a cutting board on a wooden table with a glass of beer in the background and a slice of pizza is on a wooden

**T$_{inter}$$^{1000}$:** a wooden cutting board with a pizza cut in half on top of a wooden cutting board with a glass of beer in the back of the plate

**T$_{inter}$$^{1000}$:** a wooden cutting board with two slices of pizza sitting on top of a wooden table with a glass of beer on the side of the pizza is

---

**Base:** a stop sign on a pole with a sticker attached to the stop sign on a pole with a sticker attached to the stop sign and a sticker on the pole with a sticker

**Q$_8$:** a stop sign on a pole with a street sign in the middle of the picture and a stop sign in the middle of the pole is red and white with a stop sign in the middle of the

**ToME:** a stop sign on a pole with a stop sign attached to the pole and a 3 way sign on a pole with a stop sign on it and a 3 way sign on a 3 way sign on

**T$_{intra}$$^{0.5}$:** a red stop sign on a pole with a street sign in the middle of the sign and a stop sign in the middle of the sign

**T$_{inter}$$^{100}$:** a stop sign on a metal pole with graffiti written on the side of it

**T$_{inter}$$^{1000}$:** a stop sign on a pole with a street sign in the middle of the pole and a stop sign in the middle of the pole is red and white on the pole is a stop sign with a

**T$_{inter}$$^{10000}$:** a stop sign on a pole on the side of the road

---

**Base:** the sky is clear and blue with no clouds in sight

**Q$_8$:** a large military plane sitting on top of an airport runway

**ToME:** a large group of people standing in front of a large plane on a runway at an air field with a plane on the ground in front of a large group of other planes and a large group of

**T$_{intra}$$^{0.5}$:** a group of people standing around a large plane on a runway with a blue sky in the background and a plane is on the ground and people are walking around the plane is on the ground

**T$_{inter}$$^{100}$:** a large military plane sitting on top of a grass covered field next to a crowd of people walking around it

**T$_{inter}$$^{1000}$:** a large military plane on display at an airs airs airs airs airs airs airs airs airs airs airs airs airs airs airs airs airs airs airs airs

**T$_{inter}$$^{10000}$:** a large military plane on a runway at an airs airs airs airs airs airs airs airs airs airs airs airs airs airs airs airs airs airs

Figure 5: Visualization of long-form captioning on COCO. Longer captions are generated via a length penalty of 2.0 and a maximum length of 40. Interestingly, the smaller vocabulary of $\mathcal{T}_{inter}^{100}$ possesses higher descriptiveness and coherence than $\mathcal{T}_{intra}^{0.5}$, $\mathcal{T}_{inter}^{1000}$, or $\mathcal{T}_{inter}^{10000}$ despite its lesser compression.

(including [CLS]) on different datasets. Unlike $\mathcal{T}_{intra}^{0.5}$, the sequence lengths induced by $\mathcal{T}_{inter}^{\mathcal{V}}$ are not equal. First, we compare $\mathcal{T}_{inter}^{\mathcal{V}}$ pre-processed on the in-domain dataset and ImageNet-1K (Deng et al., 2009). The in-domain dataset is represented by the training split if available. On text data, in-domain tokenizers (Gee et al., 2022; 2023a; Dagan et al., 2024; Yamaguchi et al., 2024; Minixhofer et al., 2024) have been shown to produce shorter sequences by specializing the vocabulary on the given distribution. Interestingly, we observe no such effect with image data as seen by the similar lengths between In-Domain and ImageNet-1K. Only on CelebA, do we see a slightly greater reduction with $\mathcal{T}_{inter}^{1000}$ and $\mathcal{T}_{inter}^{10000}$ pre-processed on ImageNet-1K. Second, unlike text tokenizers, decreasing compression is seen as vocabulary size increases. With text, larger vocabularies ensure that more tokens are kept as words rather than subwords. We posit that an increasing number of patches are matched to separate visual words, thus lowering the overall compression.

Having analyzed the effects on token length, we turn to the practical metrics of wattage and runtime. For online inference, savings in wattage should not incur a significant cost to runtime. In Table 2, we compare the efficiency of VWTs to $Base$, $Q_8$, and $ToME$. Note that we compute both metrics using the image encoder of CLIP or BLIP only. First, we find the wattage of VWTs to be lower than $Base$ across the datasets. On COCO, this reduction is up to 25% with $\mathcal{T}_{inter}^{100}$. Naturally, efficiency decreases as vocabulary size increases due to smaller compression. Although $Q_8$ and $ToME$ result in a lower wattage than VWTs, we note the drawbacks in runtime. In particular, $Q_8$ results in a significantly longer runtime for $Base$ as tensors need to be repeatedly quantized and dequantized. Meanwhile, $ToME$ can increase runtime by up to 100% on Waterbirds, CelebA, MetaShift, and OpenImages. We observe increases of up to 20% at most with $\mathcal{T}_{inter}^{10000}$ on the aforementioned datasets. $\mathcal{T}_{intra}^{0.5}$ displays lower runtimes than $Base$ across the datasets while $\mathcal{T}_{inter}^{100}$, $\mathcal{T}_{inter}^{1000}$, and $\mathcal{T}_{inter}^{10000}$ do so on COCO and NoCaps. We have shown how VWTs are more suitable for online inference by striking a good balance between savings in wattage and costs to runtime.

| Dataset | Model | $\mathcal{T}_{inter}^{100}$ | | $\mathcal{T}_{inter}^{1000}$ | | $\mathcal{T}_{inter}^{10000}$ | |
|---|---|---|---|---|---|---|---|
| | | In-Domain | ImageNet | In-Domain | ImageNet | In-Domain | ImageNet |
| Waterbirds | CLIP | 180 | 179 | 179 | 179 | 182 | 182 |
| CelebA | | 153 | 154 | 157 | 154 | 170 | 165 |
| MetaShift | | 176 | 173 | 175 | 174 | 180 | 180 |
| OpenImages (Com.) | | - | 171 | - | 171 | - | 176 |
| OpenImages (Rare) | | - | 169 | - | 169 | - | 174 |
| COCO | BLIP | 405 | 406 | 409 | 408 | 442 | 440 |
| NoCaps | | - | 394 | - | 396 | - | 431 |

Table 4: Token length per sample (including [CLS]). $\mathcal{T}_{inter}^{\mathcal{V}}$ of varying pre-processing data and vocabulary sizes are shown. Visual words are formed from patches after the pre-trained convolution layer of CLIP or BLIP. Unlike Table 1, poor compression is seen due to dissimilarity between patches and new visual words.

## 4.4 VWTs and Visual Performance

Another important factor in compression is the effect on model performance. In Table 3, we tabulate the performance of image classification and captioning using CLIP and BLIP, respectively. For visual recognition and subgroup robustness, we analyze performance from an average and worst-group perspective as done by Sagawa et al. (2019) and Romiti et al. (2022). First, we find the degradation in average accuracy to be small for the VWTs. The largest drop of up to 2% only is shown by $\mathcal{T}_{intra}^{0.5}$ on MetaShift. Likewise, both $Q_8$ and $ToME$ retain a high average accuracy across the datasets. Second, we observe possible improvements in subgroup robustness with VWTs. On Waterbirds and CelebA, the worst-group accuracy (WGA) with $\mathcal{T}_{intra}^{0.5}$ increases by up to 29% and 8%, respectively. Only on MetaShift do we observe a lower WGA than $Q_8$ or $ToME$ relative to *Base*. Like Gee et al. (2023b), we find compression to not always be harmful to subgroup robustness by improving the WGA at a negligible cost to overall performance.

To further validate performance, we conduct additional evaluations on the large-scale OpenImages v6 dataset for zero-shot classification. We report mean Average Precision (mAP) as done by Huang et al. (2023). Likewise, from Tables 2 and 3, we find the VWTs to be more suitable for online inference than $Q_8$ or $ToME$ in the large-scale setting while retaining comparable performance to *Base*.

For image captioning with BLIP, we evaluate our models following the setting in Li et al. (2022) by using the BLEU, CIDEr, and SPICE metrics w.r.t. the ground truth captions. On COCO, the VWTs display a higher performance than $ToME$ and are competitive with $Q_8$. On NoCaps, we see the largest degradation on the out-of-domain samples with $\mathcal{T}_{inter}^{100}$. However, overall performance of $\mathcal{T}_{intra}^{0.5}$, $\mathcal{T}_{inter}^{1000}$, and $\mathcal{T}_{inter}^{10000}$ are still higher than $ToME$. Only $Q_8$ displays a slight improvement over *Base*. Like $ToME$, the VWTs are shown to not require additional end-to-end fine-tuning for performance retention.

The generated captions by BLIP (Li et al., 2022) can be used as priors for further training of other models. As such, longer descriptive captions may be more desirable than the typical short captions associated with Internet data. In Figure 5, we visualize the long-form captions on COCO. To enable longer generations, we set the length penalty to 2.0 and double the maximum length to 40. All other generation settings are kept the same. With longer captions, the generation may degenerate into unnecessary repetitions on certain samples. Interestingly, descriptiveness and coherence improves more with $\mathcal{T}_{inter}^{100}$ than $\mathcal{T}_{intra}^{0.5}$, $\mathcal{T}_{inter}^{1000}$, or $\mathcal{T}_{inter}^{10000}$ inspite of its higher sequence compression as seen on COCO in Table 1.

## 4.5 Visualization of the Visual Words

To better understand the VWT, we visualize the patch matching of $\mathcal{T}_{intra}^{0.5}$ and $\mathcal{T}_{inter}^{100}$ in Figures 3 and 1, respectively. For the latter, we highlight in identical colors patches that are matched with one another. With $\mathcal{T}_{intra}^{0.5}$, the patches with the lowest pixel variance typically correspond to uninformative backgrounds. In most cases, the dropping of such patches continues to preserve the foreground object.

With $\mathcal{T}_{inter}^{100}$, we find that patches representing the background are more frequently grouped than those of the foreground. On Waterbirds, the merging of background patches may explain the improved robustness

| Dataset | Model | $\mathcal{T}_{inter}^{100}$ | $\mathcal{T}_{inter}^{1000}$ | $\mathcal{T}_{inter}^{10000}$ |
|---|---|---|---|---|
| Waterbirds CelebA MetaShift OpenImages (Com.) OpenImages (Rare) | CLIP | 88 | 179 | 195 |
| COCO NoCaps | BLIP | 104 | 439 | 561 |

(a) Length

| Model | Waterbirds | | CelebA | | MetaShift | | Overall ↑ | |
|---|---|---|---|---|---|---|---|---|
| | Average ↑ | Worst ↑ | Average ↑ | Worst ↑ | Average ↑ | Worst ↑ | Common ↑ | Rare ↑ |
| $\mathcal{T}_{inter}^{100}$ | 66.60 | 9.61 | 90.24 | 56.48 | 76.16 | 61.43 | 41.06 | 39.84 |
| $\mathcal{T}_{inter}^{1000}$ | 78.10 | 22.74 | 90.25 | 47.96 | 94.20 | 87.18 | 70.29 | 63.48 |
| $\mathcal{T}_{inter}^{10000}$ | 78.98 | 21.96 | 89.61 | 47.46 | 95.27 | 87.18 | 70.52 | 63.40 |

(b) CLIP (image classification and subgroup robustness)

| Model | COCO | | NoCaps | | | | | | | |
|---|---|---|---|---|---|---|---|---|---|---|
| | Karpathy ↑ | | In-Domain ↑ | | Near-Domain ↑ | | Out-of-Domain ↑ | | Overall ↑ | |
| | BLEU@4 | CIDEr | CIDEr | SPICE | CIDEr | SPICE | CIDEr | SPICE | CIDEr | SPICE |
| $\mathcal{T}_{inter}^{100}$ | 9.92 | 19.18 | 15.81 | 7.20 | 10.95 | 6.25 | 10.55 | 6.63 | 11.57 | 6.47 |
| $\mathcal{T}_{inter}^{1000}$ | 32.06 | 100.52 | 96.87 | 14.04 | 91.83 | 13.47 | 89.52 | 12.96 | 92.09 | 13.45 |
| $\mathcal{T}_{inter}^{10000}$ | 34.04 | 107.35 | 103.55 | 14.49 | 98.22 | 14.00 | 94.08 | 13.48 | 98.15 | 13.97 |

(c) BLIP (image captioning)

Table 5: Token length (including [CLS]) and performance of the *inter*-image approach with random merging. The pairwise cosine distance is initialized by sampling from a uniform distribution of $[0, 2]$. Average and worst-group accuracies degrade noticeably with $\mathcal{T}_{inter}^{100}$ except for CelebA. Performance does not change much for $\mathcal{T}_{inter}^{1000}$ and $\mathcal{T}_{inter}^{10000}$ from *Base* since barely any compression occurs.

in Table 3 by mitigating the spurious correlation with the background. On CelebA, $\mathcal{T}_{inter}^{100}$ tends to avoid matching the eyes or mouths of the individuals. We also observe that patch matching is capable of grouping similar but non-adjacent visual concepts. In certain examples of MetaShift and NoCaps, multiple cats and foods are seemingly matched together, respectively. Our analysis shows that patch matching serves as a rudimentary form of image segmentation by identifying similar visual concepts.

In Figure 4, we visualize the vocabulary of $\mathcal{T}_{inter}^{100}$ to analyze the formation of the visual words. We show patches from ImageNet-1K (i.e. pre-processing data) that are matched to each visual word using the Euclidean distance. Since visual words are centroids, patches that are matched to the same visual word belong to the same cluster. We observe that visual words depict basic features such as colors or edges. These features are also formed as an average representation of the patches that belong to each cluster. By representing basic features, visual words serve as effective anchor points to which patches can be matched to.

## 4.6 Ineffectiveness of the Embedding Space

In Table 4, we analyze the sequence compression using visual words formed in the embedding space. Instead of initializing the vocabulary by clustering patches in the pixel space, we do so with patches after the pre-trained convolution layer of CLIP or BLIP. During inference, we match the patches after the pre-trained convolution layer with these new visual words. Compared to Table 1, we observe a notable reduction in the

| Dataset | Thresh | Length | Average ↑ | Worst ↑ |
|---|---|---|---|---|
| Waterbirds | 0.2 | 97 | 76.71 | 24.09 |
| | 0.3 | 79 | 74.85 | 21.13 |
| | 0.4 | 66 | 73.35 | 16.30 |
| | 0.5 | 58 | 72.81 | 13.14 |
| CelebA | 0.2 | 66 | 89.44 | 56.48 |
| | 0.3 | 55 | 88.85 | 58.70 |
| | 0.4 | 50 | 88.31 | 57.04 |
| | 0.5 | 48 | 88.21 | 55.74 |
| MetaShift | 0.2 | 84 | 86.16 | 77.46 |
| | 0.3 | 69 | 81.16 | 67.77 |
| | 0.4 | 60 | 77.73 | 60.65 |
| | 0.5 | 55 | 74.75 | 56.08 |

Table 6: Token length (including [CLS]) and performance using $\mathcal{T}_{inter}^{100}$ with varying similarity thresholds. Higher thresholds can reduce sequences to 48 tokens on CelebA. Greater compression results in higher degradation in performance, particularly in subgroup robustness.

compression irrespective of pre-processing data and vocabulary size. In Tang et al. (2022), embeddings are shown to become progressively more similar to one another due to self-attention. As such, it is unsurprising that matching patches and visual words in the initial embedding space is found to be ineffective.

### 4.7 Random Merging of Tokens

In Table 5, we study the effects of randomly merging the tokens. We initialize the pairwise cosine distance by sampling from a uniform distribution of $[0, 2]$. First, we find the token length (including [CLS]) to differ noticeably from Table 1. For $\mathcal{T}_{inter}^{100}$, sequences are further reduced across the datasets. Conversely, $\mathcal{T}_{inter}^{1000}$ and $\mathcal{T}_{inter}^{10000}$ display little compression. Second, performance is shown to change from Table 3. We observe a significant degradation in average and worst-group accuracies with $\mathcal{T}_{inter}^{100}$ on Waterbirds and MetaShift. For $\mathcal{T}_{inter}^{1000}$ and $\mathcal{T}_{inter}^{10000}$, performance does not shift much from *Base* as barely any compression occurs. With random merging, the captions are shown to deviate completely from those of *Base* in Figure 6, thus further demonstrating that visual words are an effective criterion for grouping patches as shown in Figure 1.

### 4.8 Ablation of the Similarity Threshold

We have shown how the *inter*-image approach can improve the efficiency of online inference. To better understand their limitations, we ablate the threshold of $\mathcal{T}_{inter}^{100}$ by setting it to values of $\{0.2, 0.3, 0.4, 0.5\}$ in Table 6. We seek to determine if exploiting higher thresholds for increased compression is a viable strategy. Naturally, we observe a reduction in performance as increasingly dissimilar patches are merged. For Waterbirds and MetaShift, the WGA degrades more significantly than the average, especially with the former. Interestingly, average accuracy remains relatively unchanged while WGA improves significantly on CelebA irrespective of similarity threshold. We posit that at higher thresholds, the merging of core features represented by the foreground object results in the reduced performance of Waterbirds and MetaShift.

### 4.9 When to use the *intra*-image or *inter*-image approach?

Concerning the choice between the *intra*-image or *inter*-image approach, we posit that when global information is required, dropping tokens may be more advantageous by removing unnecessary noise from the visual input. On the other hand, when the task necessitates local information (e.g. long-form captioning), merging tokens may better preserve the visual concepts. For example, the image on the first row of Figure 7 of Appendix A.2 shows that $\mathcal{T}_{intra}^{0.5}$ removes the mountainous background, thus leading to the absence of

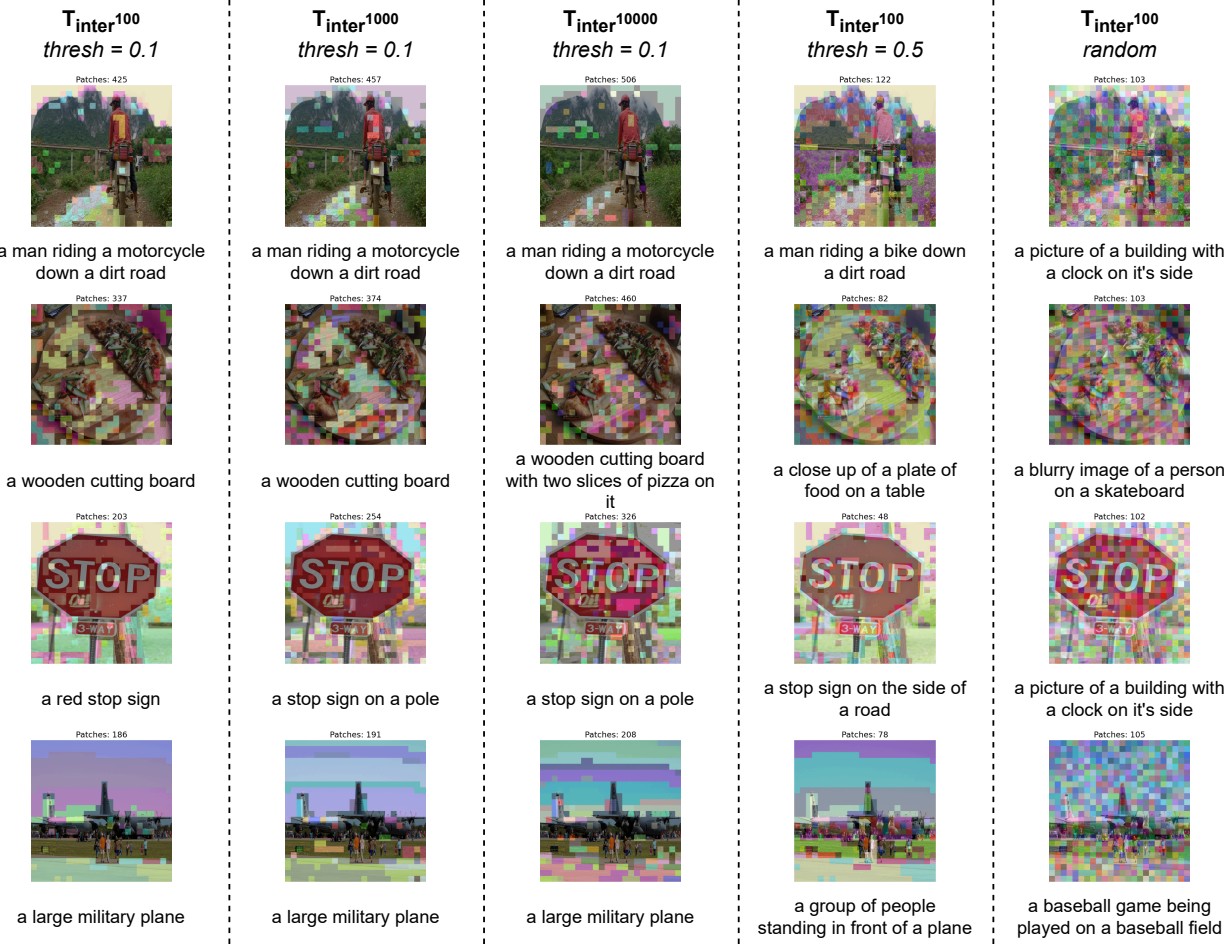

Figure 6: Visualization of image captions on COCO by the *inter*-image approach. The generated captions are shown to deviate more when increasing the similarity threshold than when reducing the vocabulary size. With random matching, the model begins to completely misunderstand the image.

"mountain" in the long-form caption of Figure 5. Potentially, combining both approaches may maximize compression while preserving information by merging the mountains and dropping the bushes.

## 5  Conclusion

In this work, we set out to define a training-free tokenization for ViTs that lowers wattage while balancing costs to runtime and performance. In online scenarios, we have shown empirically that our *intra*-image and *inter*-image approaches are stronger than 8-bit quantization and token merging for image classification and captioning. Analysis on large-scale classification further validates the viability of our method while long-form captioning shows its potential for improving descriptiveness and coherence. Qualitatively, we observe how the criterion of the *intra*-image approach typically corresponds to the background while that of the *inter*-image approach groups analogous visual concepts based on visual words that represent basic features. As a future work, additional research could explore combining both the *intra*-image and *inter*-image approaches, potentially improving further the performance and efficiency of the VWT.

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

## A  Further Details

### A.1  Datasets

Here, we detail the task of each dataset for subgroup robustness. In Table 7, we also tabulate the labels and attributes that define each subgroup along with their sample sizes.

**Waterbirds.**  Given an image of a bird, the task is to predict whether it is a waterbird or landbird (Wah et al., 2011). Following Sagawa et al. (2019), the attribute is the background that the bird is on. We use the same dataset splits as Liu et al. (2021).

**CelebA.**  Given an image of a person, the task is to predict whether their hair color is blond or not (Liu et al., 2015). Following Sagawa et al. (2019), the attribute is the binary gender of the person. We use the same dataset splits as Liu et al. (2021).

**MetaShift.**  Given an image of an animal, the task is to predict whether it is a dog or cat (Liang & Zou, 2022). Following Liang & Zou (2022), the attribute is the environment that the dog or cat is in. We use the same dataset splits as Yang et al. (2023).

### A.2  Dropping Ratios

We tabulate the results with additional dropping ratios ($\{0.25, 0.33, 0.7\}$) used by Minderer et al. (2024) in Table 8. We observe a natural degradation as the ratio increases with performance dropping steeply at 0.7.

### A.3  Code Implementation

We provide our code as part of the supplementary materials. Code for the VWT can be found in *utils/vwt.py*.

# B Supplementary Experiments

## B.1 Random Dropping of Tokens

In Table 9, we provide further analysis on the *intra*-image approach by randomly dropping tokens. This is somewhat similar to Simoulin et al. (2024) with the exclusion of any fine-tuning. First, unlike CelebA, we observe a noticeably degradation in worst-group accuracies on Waterbirds and MetaShift. We hypothesize that this stems from the target label being spuriously correlated with the background (Waterbirds, MetaShift) and not gender (CelebA), which is distinctly separable from the foreground object as shown in Figure 3 and Table 7. Hence, random dropping is beneficial to the subgroup robustness of CelebA by reducing the gender features of the individuals. Second, we find performance (except $\mathcal{T}_{intra}^{0.7}$) to be slightly lower or equivalent on OpenImages and the remaining datasets, respectively. We attribute this to cases where the *intra*-image approach misidentifies the foreground object as irrelevant information as shown in Figure 3 with the removal of the "airplane" (top image) and "building" (bottom image). However, in general, the variance of the individual patches is an effective heuristic for robustly compressing redundant information within the image.

## B.2 VWTs with Quantization

We designed the VWT to be a training-free approach to efficient online inference. As such, we exclude any form of fine-tuning (Lin et al., 2024; Hao et al., 2024; Simoulin et al., 2024) (e.g. knowledge distillation (Tian et al., 2019; Han et al., 2024; Son et al., 2021)) for performance recovery. However, should it be desired, our method may be used alongside network compression (Tian et al., 2019; Han et al., 2024; Son et al., 2021) (i.e. minimizing the Kullback–Leibler divergence between the softened outputs of the larger teacher and smaller student), memory-efficient fine-tuning (Hao et al., 2024; Simoulin et al., 2024), and other compression techniques (e.g. quantization) as the VWT only influences the initial input sequence to the vision transformer. In Table 10, we provide additional scores when VWTs are combined with 8-bit quantization. We find the performance to be similar to those in Table 3, thereby showing that VWTs are highly compatible with other compression approaches as mentioned above.

## B.3 Fairness of the Tokenization

Text tokenizers are known to induce unfairness between languages that raises compute costs, particularly for minority languages (Petrov et al., 2024; Ali et al., 2023). We seek to analyze if similar effects exist with VWTs. In Table 11, we show the breakdown in token length (including [CLS]) and accuracy (w.r.t *Base*) by subgroup. First, we observe a notable difference in compression between the subgroups of Waterbirds. With $\mathcal{T}_{inter}^{100}$, sequences might differ by up to 39 tokens as seen with subgroups 0 and 3. Smaller discrepancies are displayed on CelebA and MetaShift except for $\mathcal{T}_{inter}^{100}$ on the former. Second, we find compression to not affect all subgroups equally. Accuracy improves on certain subgroups and degrades on others. A stronger sequence compression does not correlate with a larger change in performance.

## B.4 Sparsity of the Vocabulary

To better understand the utilization of the visual words, we plot the probability distribution of the matches in Figure 8. Regardless of the dataset, we find that certain visual words are matched more frequently than others, thus leading to a large skew in the distributions. Greater sparsity is also displayed by larger vocabularies as many visual words remain unused across datasets. As such, the pruning of unmatched visual words may be applied to achieve a more efficient vocabulary size after the PRE-PROCESS step.

## B.5 Ablation of the Batch Size

In Figure 9, we seek to better understand the effectiveness of VWTs for **offline inference** where batch sizes are greater than 1. Using batches of $\{4, 8, 16, 32\}$, we compare $\mathcal{T}_{intra}^{0.5}$ and $\mathcal{T}_{inter}^{1000}$ to *Base* and *ToME*. First, we find that as batch size increases, the wattage gap with *Base* decreases. Both VWTs and *ToME* show little energy savings at larger batches (i.e. 16 and 32), particularly with $\mathcal{T}_{inter}^{1000}$. Second, we observe different runtime trends between VWTs and *ToME*. As batches grow, $\mathcal{T}_{inter}^{1000}$ continues to show similar runtimes to

*Base.* Meanwhile, $\mathcal{T}^{0.5}_{intra}$ and $ToME$ display lower runtimes with the exception of batch size 4 for the latter. Hence, the *inter*-image approach does not confer the same degree of efficiency during batching.

| Dataset | Subgroup | Label | Attribute | Training | Validation | Test |
|---------|----------|-------|-----------|----------|------------|------|
| Waterbirds | 0 | 0 (landbird) | 0 (on land) | 3498 | 467 | 2255 |
| | 1 | 0 (landbird) | 1 (on water) | 184 | 466 | 2255 |
| | 2 | 1 (waterbird) | 0 (on land) | 56 | 133 | 642 |
| | 3 | 1 (waterbird) | 1 (on water) | 1057 | 133 | 642 |
| CelebA | 0 | 0 (non-blond) | 0 (woman) | 71629 | 8535 | 9767 |
| | 1 | 0 (non-blond) | 1 (man) | 66874 | 8276 | 7535 |
| | 2 | 1 (blond) | 0 (woman) | 22880 | 2874 | 2480 |
| | 3 | 1 (blond) | 1 (man) | 1387 | 182 | 180 |
| MetaShift | 0 | 0 (dog) | 0 (outdoor) | 784 | 127 | 273 |
| | 1 | 0 (dog) | 1 (indoor) | 507 | 75 | 191 |
| | 2 | 1 (cat) | 0 (outdoor) | 196 | 33 | 65 |
| | 3 | 1 (cat) | 1 (indoor) | 789 | 114 | 345 |

Table 7: Defined subgroups in Waterbirds, CelebA, and MetaShift.

| Dataset | Model | $\mathcal{T}^{0.25}_{intra}$ | $\mathcal{T}^{0.33}_{intra}$ | $\mathcal{T}^{0.7}_{intra}$ |
|---------|-------|------|------|------|
| Waterbirds CelebA MetaShift OpenImages (Com.) OpenImages (Rare) | CLIP | 148 | 132 | 59 |
| COCO NoCaps | BLIP | 433 | 386 | 173 |

(a) Length

| Model | Waterbirds | | CelebA | | MetaShift | | OpenImages | |
|-------|------------|------|--------|------|-----------|------|------------|------|
| | Average ↑ | Worst ↑ | Average ↑ | Worst ↑ | Average ↑ | Worst ↑ | Common ↑ | Rare ↑ |
| $\mathcal{T}^{0.25}_{intra}$ | 78.31 | 26.69 | 89.46 | 48.47 | 94.70 | 87.69 | 69.79 | 63.37 |
| $\mathcal{T}^{0.33}_{intra}$ | 77.72 | 27.78 | 89.31 | 48.83 | 94.62 | 87.18 | 69.11 | 62.75 |
| $\mathcal{T}^{0.7}_{intra}$ | 73.27 | 18.80 | 89.19 | 45.37 | 82.61 | 63.59 | 46.83 | 42.68 |

(b) CLIP (image classification and subgroup robustness)

| Model | COCO | | NoCaps | | | | | | | |
|-------|------|------|--------|------|------|------|------|------|------|------|
| | Karpathy ↑ | | In-Domain ↑ | | Near-Domain ↑ | | Out-of-Domain ↑ | | Overall ↑ | |
| | BLEU@4 | CIDEr | CIDEr | SPICE | CIDEr | SPICE | CIDEr | SPICE | CIDEr | SPICE |
| $\mathcal{T}^{0.25}_{intra}$ | 33.92 | 107.47 | 103.59 | 14.65 | 98.39 | 14.07 | 94.47 | 13.59 | 98.34 | 14.06 |
| $\mathcal{T}^{0.33}_{intra}$ | 33.59 | 106.25 | 103.19 | 14.65 | 97.40 | 13.95 | 92.66 | 13.41 | 97.27 | 13.95 |
| $\mathcal{T}^{0.7}_{intra}$ | 29.64 | 93.20 | 89.89 | 13.53 | 85.04 | 12.82 | 80.81 | 12.31 | 84.88 | 12.82 |

(c) BLIP (image captioning)

Table 8: Token length (including [CLS]) and performance of the *intra*-image approach with varying dropping ratios. Performance degrades naturally as the ratio increases before falling steeply at 0.7.

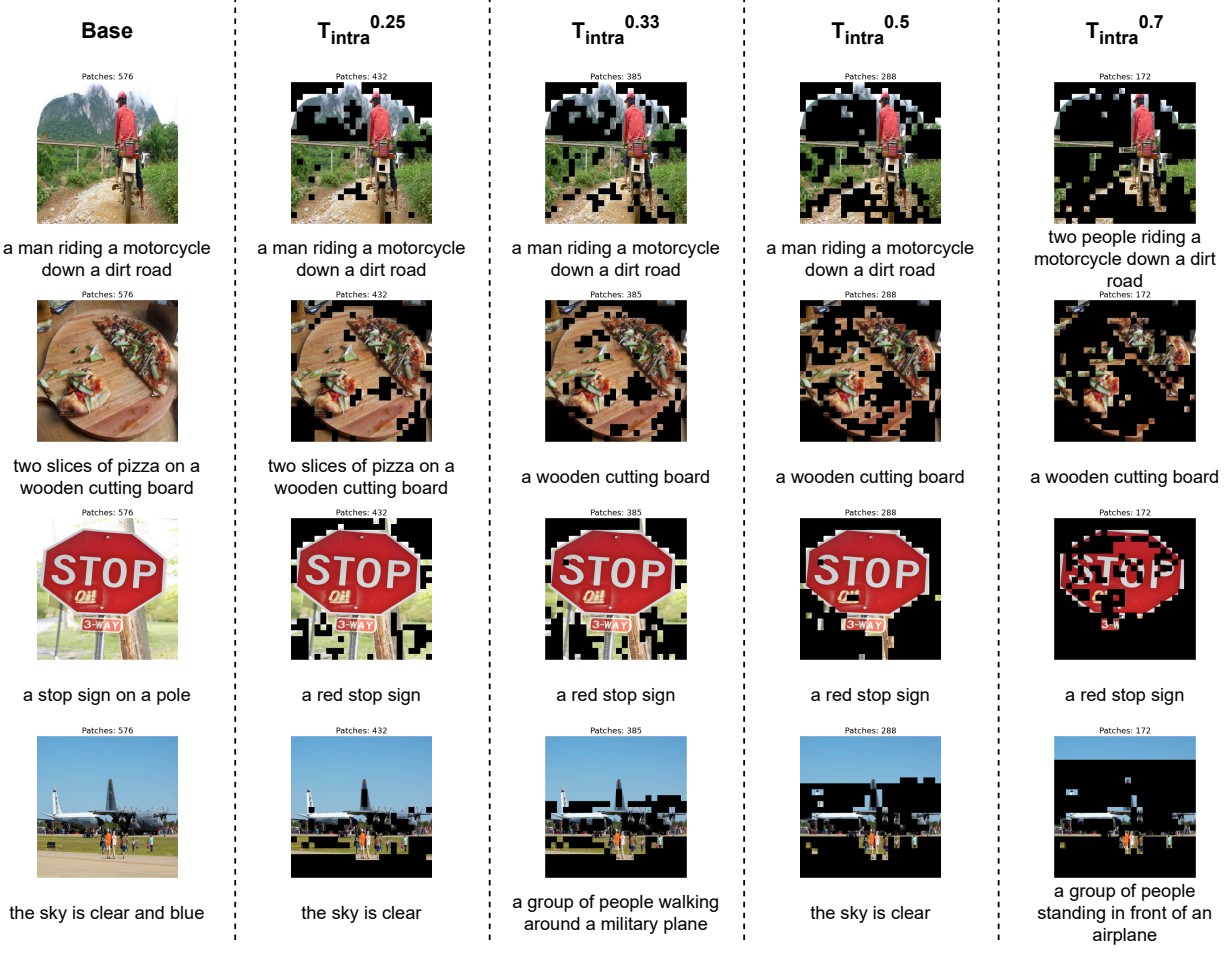

Figure 7: Visualization of image captions on COCO by the *intra*-image approach. The generated captions to not deviate significantly from those of *Base* with dropping ratios of up to 0.5.

| Model | Waterbirds | | CelebA | | MetaShift | | OpenImages | |
|---|---|---|---|---|---|---|---|---|
| | Average ↑ | Worst ↑ | Average ↑ | Worst ↑ | Average ↑ | Worst ↑ | Common ↑ | Rare ↑ |
| $\mathcal{T}_{intra}^{0.25}$ | 76.90 | 21.50 | 89.88 | 48.75 | 94.39 | 85.13 | 70.00 | 63.38 |
| $\mathcal{T}_{intra}^{0.33}$ | 75.72 | 20.51 | 90.10 | 48.89 | 93.44 | 84.62 | 69.66 | 63.06 |
| $\mathcal{T}_{intra}^{0.5}$ | 72.38 | 19.52 | 90.56 | 51.85 | 91.61 | 81.93 | 67.59 | 61.61 |
| $\mathcal{T}_{intra}^{0.7}$ | 66.78 | 11.58 | 90.24 | 61.48 | 84.29 | 69.73 | 52.72 | 49.86 |

(a) CLIP (image classification and subgroup robustness)

| Model | COCO | | NoCaps | | | | | | | |
|---|---|---|---|---|---|---|---|---|---|---|
| | Karpathy ↑ | | In-Domain ↑ | | Near-Domain ↑ | | Out-of-Domain ↑ | | Overall ↑ | |
| | BLEU@4 | CIDEr | CIDEr | SPICE | CIDEr | SPICE | CIDEr | SPICE | CIDEr | SPICE |
| $\mathcal{T}_{intra}^{0.25}$ | 33.89 | 106.59 | 103.70 | 14.39 | 98.02 | 14.01 | 93.79 | 13.24 | 97.98 | 13.92 |
| $\mathcal{T}_{intra}^{0.33}$ | 33.78 | 106.57 | 102.44 | 14.35 | 98.15 | 13.98 | 93.04 | 13.21 | 97.73 | 13.88 |
| $\mathcal{T}_{intra}^{0.5}$ | 33.32 | 104.65 | 100.70 | 14.06 | 95.11 | 13.72 | 92.25 | 13.22 | 95.33 | 13.67 |
| $\mathcal{T}_{intra}^{0.7}$ | 31.15 | 97.18 | 95.02 | 13.57 | 86.83 | 13.01 | 86.43 | 12.52 | 87.93 | 13.00 |

(b) BLIP (image captioning)

Table 9: Token length (including [CLS]) and performance of the *intra*-image approach with random dropping. Unlike CelebA, the subgroup robustness degrades noticeably on Waterbirds and MetaShift. Performance is slightly lower on OpenImages or equivalent on the remaining datasets except for $\mathcal{T}_{intra}^{0.7}$.

| Model | Waterbirds | | CelebA | | MetaShift | | OpenImages | |
|---|---|---|---|---|---|---|---|---|
| | Average ↑ | Worst ↑ | Average ↑ | Worst ↑ | Average ↑ | Worst ↑ | Common ↑ | Rare ↑ |
| $\mathcal{T}_{intra}^{0.5} + Q_8$ | 76.31 | 33.65 | 89.43 | 52.41 | 93.14 | 87.78 | 65.29 | 59.27 |
| $\mathcal{T}_{inter}^{100} + Q_8$ | 78.88 | 27.57 | 90.17 | 54.44 | 90.58 | 83.72 | 62.75 | 57.99 |
| $\mathcal{T}_{inter}^{1000} + Q_8$ | 80.01 | 25.60 | 91.05 | 50.93 | 93.67 | 86.67 | 66.15 | 60.48 |
| $\mathcal{T}_{inter}^{10000} + Q_8$ | 80.28 | 25.29 | 90.26 | 47.96 | 94.58 | 86.67 | 69.07 | 62.37 |

(a) CLIP (image classification and subgroup robustness)

| Model | COCO | | NoCaps | | | | | | | |
|---|---|---|---|---|---|---|---|---|---|---|
| | Karpathy ↑ | | In-Domain ↑ | | Near-Domain ↑ | | Out-of-Domain ↑ | | Overall ↑ | |
| | BLEU@4 | CIDEr | CIDEr | SPICE | CIDEr | SPICE | CIDEr | SPICE | CIDEr | SPICE |
| $\mathcal{T}_{intra}^{0.5} + Q_8$ | 32.45 | 103.18 | 101.98 | 14.46 | 94.93 | 13.74 | 87.13 | 12.78 | 94.36 | 13.66 |
| $\mathcal{T}_{inter}^{100} + Q_8$ | 30.88 | 97.48 | 95.97 | 13.94 | 90.27 | 13.26 | 80.02 | 12.39 | 89.01 | 13.19 |
| $\mathcal{T}_{inter}^{1000} + Q_8$ | 31.72 | 100.23 | 99.69 | 14.02 | 94.35 | 13.49 | 85.69 | 12.57 | 93.36 | 13.39 |
| $\mathcal{T}_{inter}^{10000} + Q_8$ | 32.98 | 104.15 | 101.77 | 14.34 | 98.75 | 14.02 | 91.55 | 13.21 | 97.72 | 13.91 |

(b) BLIP (image captioning)

Table 10: Zero-shot (training-free) image classification (CLIP), subgroup robustness (CLIP) and captioning (BLIP). The VWTs are applied jointly with 8-bit quantization. Performance is shown to be similar to those with VWTs only, thereby displaying the mutual compatibility of VWTs with other compression techniques.

| Model | Subgroup | Waterbirds | | CelebA | | MetaShift | |
|---|---|---|---|---|---|---|---|
| | | **Length** | **Δ Accuracy** | **Length** | **Δ Accuracy** | **Length** | **Δ Accuracy** |
| *Base* | 0 | | 98.89 | | 95.63 | | 98.54 |
| | 1 | 197 | 82.45 | 197 | 96.23 | 197 | 93.19 |
| | 2 | | 21.86 | | 48.67 | | 87.69 |
| | 3 | | 54.73 | | 50.00 | | 95.36 |
| $\mathcal{T}_{inter}^{100}$ | 0 | 144 | -0.67 | 85 | -3.25 | 118 | -0.74 |
| | 1 | 106 | -3.82 | 88 | -0.13 | 110 | -6.18 |
| | 2 | 140 | 19.00 | 98 | 32.62 | 119 | -2.92 |
| | 3 | 105 | 6.17 | 99 | 6.30 | 100 | -9.02 |
| $\mathcal{T}_{inter}^{1000}$ | 0 | 160 | -0.40 | 119 | -0.46 | 138 | -0.25 |
| | 1 | 129 | -0.22 | 117 | 0.74 | 138 | -2.06 |
| | 2 | 160 | 9.03 | 124 | 18.92 | 143 | -1.17 |
| | 3 | 129 | 2.18 | 124 | -4.07 | 131 | -2.13 |
| $\mathcal{T}_{inter}^{10000}$ | 0 | 180 | -0.09 | 158 | 0.25 | 163 | -0.37 |
| | 1 | 156 | 1.34 | 151 | 0.66 | 167 | 0.19 |
| | 2 | 181 | 4.75 | 158 | 2.96 | 169 | -1.75 |
| | 3 | 158 | 1.71 | 156 | -6.30 | 163 | -0.81 |

Table 11: Distribution of token length (including [CLS]) and accuracy (w.r.t. *Base*) by subgroup. $\mathcal{T}_{inter}^{\mathcal{V}}$ of varying vocabulary sizes are shown. Like text tokenizers, VWTs may induce unequal token lengths as seen with $\mathcal{T}_{inter}^{100}$ on Waterbirds. Performance is also affected unequally as a stronger sequence compression does not correlate with a greater improvement or degradation in accuracy.

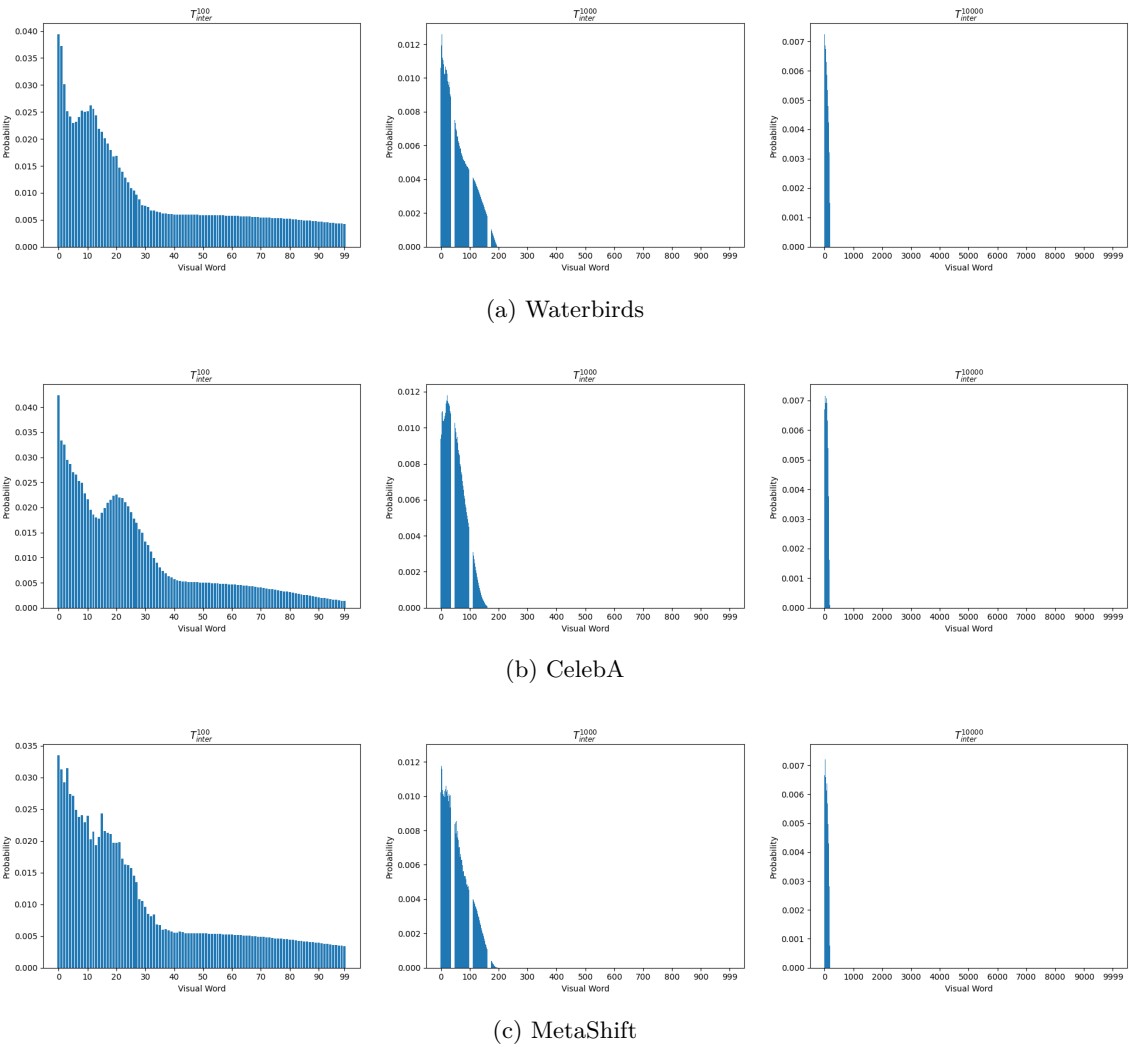

Figure 8: Probability distribution of the matched visual words. $\mathcal{T}^{\mathcal{V}}_{inter}$ of varying vocabulary sizes are shown. The probability distribution exhibits a large skew irrespective of the dataset as certain visual words are matched more frequently than others. Larger vocabularies display greater sparsity as the many visual words that remain unmatched may be pruned for a more efficient vocabulary size.

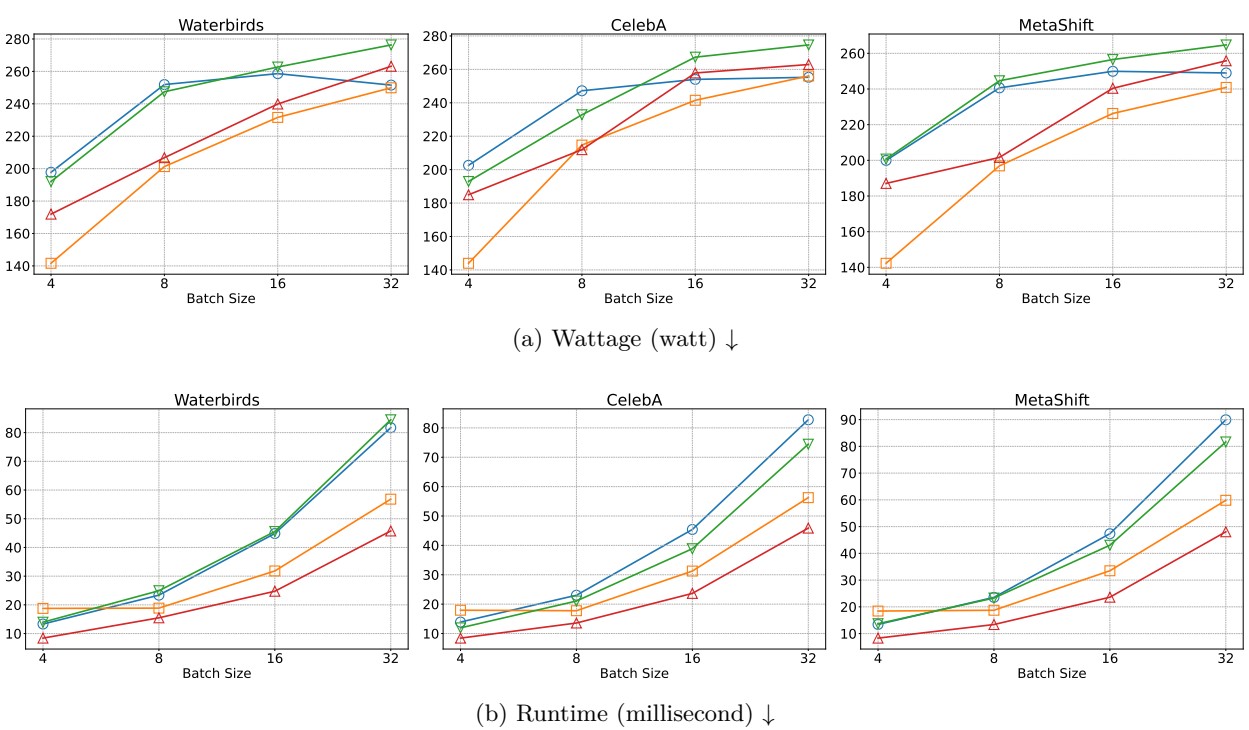

(a) Wattage (watt) ↓

(b) Runtime (millisecond) ↓

Figure 9: Wattage and runtime with varying batch sizes. $\mathcal{T}_{intra}^{0.5}$ (△) and $\mathcal{T}_{inter}^{1000}$ (▽) are compared to *Base* (◯) and *ToME* (□). Both VWTs and *ToME* show little energy savings relative to *Base* as the batch size grows, particularly with $\mathcal{T}_{inter}^{1000}$. Runtime also increases more for $\mathcal{T}_{inter}^{1000}$ except with smaller batches.

