# OpenReview forum: "Visual-Word Tokenizer: Beyond Fixed Sets of Tokens in Vision Transformers"
_TMLR — Rejected by TMLR_

### Review · Reviewer_bsiy · 2025-04-28

**Summary Of Contributions:**

The authors argue that existing compression techniques require additional end-to-end fine-tuning or incur a significant drawback to runtime. In this sense, the authors propose the Visual Word Tokenizer (VWT), which is training-free to reduce energy cost while retaining performance and runtime.

**Audience:**

Yes

**Claims And Evidence:**

No

**Requested Changes:**

Please address my above concerns.

**Strengths And Weaknesses:**

Strength: The perspective of this paper is interesting. The authors consider images as visual subwords and classify them into frequent and infrequent ones. This design reduces the power consumption.

Weakness:
1. The comparison in this paper is not comprehensive. And the starting point of saying power reduction is somehow misleading. In [1-3], especially [3], even random selection of transformer tokens can result in a significantly lower memory consumption (in the authors' case, power reduction). The significance of introducing visual subwords as guidance thus becomes weak.

2. To solve 1, the authors need to discuss in detail the related topics, such as network compression [4-6] and memory-efficient fine-tuning, to re-emphasize their novelty.

[1] Not All Tokens Are What You Need for Pretraining

[2] MEFT: Memory-Efficient Fine-Tuning through Sparse Adapter

[3] Memory-Efficient Fine-Tuning of Transformers via Token Selection

[4] Contrastive representation distillation

[5] AMD: Automatic Multi-step Distillation of Large-scale Vision Models

[6] Densely guided knowledge distillation using multiple teacher assistants

---

> ### Author Response · Authors · 2025-05-13
> **Rebuttal to Reviewer bsiy**
>
> Hi, thank you for the review. We seek to address your concerns below.
>
> First, as stated in our paper, the Visual-Word Tokenizer (VWT) is a **training-free** approach to efficient online inference. As such, we exclude any fine-tuning [1-3] (e.g. knowledge distillation [4-6]) for performance recovery. However, should it be desired, our method may be used alongside network compression [4-6] (i.e. minimizing the Kullback–Leibler divergence between the softened outputs of the larger teacher and smaller student), memory-efficient fine-tuning [2-3], and other compression techniques (e.g. quantization) as the VWT only influences the initial input sequence to the vision transformer. In the Table below, we provide additional scores when VWTs are combined with 8-bit quantization ($Q_{8}$). We find the performance to be similar to those in Table 3 of our paper (included below for convenience), thereby showing that VWTs are highly compatible with other compression approaches as mentioned above.
>
> | Model | Waterbirds (Average) ↑ | Waterbirds (Worst) ↑ | CelebA (Average) ↑ | CelebA (Worst) ↑ | MetaShift (Average) ↑ | MetaShift (Worst) ↑ | OpenImages (Common) ↑ | OpenImages (Rare) ↑ |
> | --- | --- | --- | --- | --- | --- | --- | --- | --- |
> | $T_{intra}^{0.5}$ | 75.56 | 31.00 | 89.27 | 50.93 | 92.94 | 86.15 | 65.52 | 59.73 |
> | $T_{intra}^{0.5} + Q_{8}$ | 76.31 | 33.65 | 89.43 | 52.41 | 93.14 | 87.78 | 65.29 | 59.27 |
> | $T_{inter}^{100}$ | 78.41 | 26.01 | 90.04 | 53.15 | 90.24 | 84.28 | 62.90 | 58.10 |
> | $T_{inter}^{100} + Q_{8}$ | 78.88 | 27.57 | 90.17 | 54.44 | 90.58 | 83.72 | 62.75 | 57.99 |
> | $T_{inter}^{1000}$ | 79.19 | 23.83 | 90.79 | 47.96 | 93.94 | 86.67 | 66.15 | 60.56 |
> | $T_{inter}^{1000} + Q_{8}$ | 80.01 | 25.60 | 91.05 | 50.93 | 93.67 | 86.67 | 66.15 | 60.48 |
> | $T_{inter}^{10000}$ | 79.68 | 22.90 | 90.12 | 46.85 | 94.81 | 86.15 | 69.03 | 62.50 |
> | $T_{inter}^{10000} + Q_{8}$ | 80.28 | 25.29 | 90.26 | 47.96 | 94.58 | 86.67 | 69.07 | 62.37 |
>
> | Model | COCO (BLEU@4) ↑ | COCO (CIDEr) ↑ | NoCaps (Overall - CIDEr) ↑ | NoCaps (Overall - SPICE) ↑ |
> | --- | --- | --- | --- | --- |
> | $T_{intra}^{0.5}$ | 32.80 | 104.37 | 94.07 | 13.68 |
> | $T_{intra}^{0.5} + Q_{8}$ | 32.45 | 103.18 | 94.36 | 13.66 |
> | $T_{inter}^{100}$ | 31.10 | 97.70 | 89.02 | 13.08 |
> | $T_{inter}^{100} + Q_{8}$ | 30.88 | 97.48 | 89.01 | 13.19 |
> | $T_{inter}^{1000}$ | 32.12 | 101.79 | 92.89 | 13.39 |
> | $T_{inter}^{1000} + Q_{8}$ | 31.72 | 100.23 | 93.36 | 13.39 |
> | $T_{inter}^{10000}$ | 33.18 | 105.08 | 97.40 | 13.88 |
> | $T_{inter}^{10000} + Q_{8}$ | 32.98 | 104.15 | 97.72 | 13.91 |
>
> Second, we already include an analysis on **randomly merging** tokens for the *inter*-image approach in Table 5 of our paper (included below for convenience). Performance is shown to collapse ($T_{inter}^{100}$) or  not change significantly ($T_{inter}^{1000}$, $T_{inter}^{10000}$) due to the merging of dissimilar tokens or lack of compression, respectively. As such, the visual words are necessary for correctly identifying similar tokens to merge.
>
> | Model | Waterbirds (Average) ↑ | Waterbirds (Worst) ↑ | CelebA (Average) ↑ | CelebA (Worst) ↑ | MetaShift (Average) ↑ | MetaShift (Worst) ↑ | OpenImages (Common) ↑ | OpenImages (Rare) ↑ |
> | --- | --- | --- | --- | --- | --- | --- | --- | --- |
> | $T_{inter}^{100}$ (VWT) | 78.41 | 26.01 | 90.04 | 53.15 | 90.24 | 84.28 | 62.90 | 58.10 |
> | $T_{inter}^{100}$ (random) | 66.60 | 9.61 | 90.24 | 56.48 | 76.16 | 61.43 | 41.06 | 39.84 |
> | $T_{inter}^{1000}$ (VWT) | 79.19 | 23.83 | 90.79 | 47.96 | 93.94 | 86.67 | 66.15 | 60.56 |
> | $T_{inter}^{1000}$ (random) | 78.10 | 22.74 | 90.25 | 47.96 | 94.20 | 87.18 | 70.29 | 63.48 |
> | $T_{inter}^{10000}$ (VWT) | 79.68 | 22.90 | 90.12 | 46.85 | 94.81 | 86.15 | 69.03 | 62.50 |
> | $T_{inter}^{10000}$ (random) | 78.98 | 21.96 | 89.61 | 47.46 | 95.27 | 87.18 | 70.52 | 63.40 |
>
> | Model | COCO (BLEU@4) ↑ | COCO (CIDEr) ↑ | NoCaps (Overall - CIDEr) ↑ | NoCaps (Overall - SPICE) ↑ |
> | --- | --- | --- | --- | --- |
> | $T_{inter}^{100}$ (VWT) | 31.10 | 97.70 | 89.02 | 13.08 |
> | $T_{inter}^{100}$ (random) | 9.92 | 19.18 | 11.57 | 6.47 |
> | $T_{inter}^{1000}$ (VWT) | 32.12 | 101.79 | 92.89 | 13.39 |
> | $T_{inter}^{1000}$ (random) | 32.06 | 100.52 | 92.09 | 13.45 |
> | $T_{inter}^{10000}$ (VWT) | 33.18 | 105.08 | 97.40 | 13.88 |
> | $T_{inter}^{10000}$ (random) | 34.04 | 107.35 | 98.15 | 13.97 |

---

> ### Author Response · Authors · 2025-05-13
> **Rebuttal to Reviewer bsiy**
>
> In the Table below, we provide additional scores for the *intra*-image approach by **randomly dropping** tokens. This is somewhat similar to [3] with the exclusion of any fine-tuning. On Waterbirds and MetaShift, performance degrades noticeably, particularly the worst-group accuracy (i.e. subgroup robustness). This is due to the spurious attribute being the background as shown in Figure 3 and Table 7 (in the Appendix) of our of paper and is not targeted when tokens are randomly dropped. Meanwhile, the scores on OpenImages and the remaining datasets are slightly higher or equivalent, respectively. We attribute this to cases where the *intra*-image approach misidentifies the foreground object as irrelevant information as shown in Figure 3 with the removal of the “airplane” (top image) and “building” (bottom image). However, in general, the variance of the individual patches is an effective heuristic for robustly compressing redundant information within the image.
>
> | Model | Waterbirds (Average) ↑ | Waterbirds (Worst) ↑ | CelebA (Average) ↑ | CelebA (Worst) ↑ | MetaShift (Average) ↑ | MetaShift (Worst) ↑ | OpenImages (Common) ↑ | OpenImages (Rare) ↑ |
> | --- | --- | --- | --- | --- | --- | --- | --- | --- |
> | $T_{intra}^{0.5}$ (VWT) | 75.56 | 31.00 | 89.27 | 50.93 | 92.94 | 86.15 | 65.52 | 59.73 |
> | $T_{intra}^{0.5}$ (random) | 72.38 | 19.52 | 90.56 | 51.85 | 91.61 | 81.93 | 67.59 | 61.61 |
>
> | Model | COCO (BLEU@4) ↑ | COCO (CIDEr) ↑ | NoCaps (Overall - CIDEr) ↑ | NoCaps (Overall - SPICE) ↑ |
> | --- | --- | --- | --- | --- |
> | $T_{intra}^{0.5}$ (VWT) | 32.80 | 104.37 | 94.07 | 13.68 |
> | $T_{intra}^{0.5}$ (random) | 33.32 | 104.56 | 95.33 | 13.67 |
>
> We will include these additional discussions in the revised version of our paper.
>
> **References**
>
> [1] Not All Tokens Are What You Need for Pretraining
>
> [2] MEFT: Memory-Efficient Fine-Tuning through Sparse Adapter
>
> [3] Memory-Efficient Fine-Tuning of Transformers via Token Selection
>
> [4] Contrastive representation distillation
>
> [5] AMD: Automatic Multi-step Distillation of Large-scale Vision Models
>
> [6] Densely guided knowledge distillation using multiple teacher assistants

---

### Review · Reviewer_ZW4V · 2025-05-07

**Summary Of Contributions:**

The submission introduces the Visual Word Tokenizer (VWT), a training-free method designed to reduce the energy consumption of vision transformers (ViTs) while largely preserving performance and inference runtime.  The core idea is to group frequently occurring visual subwords (image patches) into "visual words," while less frequent patches are kept as individual subwords.  This grouping is achieved by leveraging either intra-image statistics (specifically, pixel variance to identify and drop uniform areas ) or inter-image statistics (using k-means clustering on image patches to form a vocabulary of visual words for merging similar patches ). The authors report experimental results demonstrating up to a 25% reduction in wattage with at most a 20% increase in runtime.

**Audience:**

Yes

**Claims And Evidence:**

Yes

**Requested Changes:**

- Clarify Novelty: Explicitly differentiate the inter-image VWT (especially the clustering for token pruning/merging) from existing BoVW models and related vision tokenization methods. What is the core new contribution beyond applying these ideas?

- Detail "Visual Word" Process: Specify the exact patch features used for k-means clustering in the "image space". Elaborate on how similarity (e.g., cosine distance) is computed with these patch representations.

**Strengths And Weaknesses:**

# Strengths
- Training-Free: Easy to apply to pre-trained models.
- Energy Savings: Offers notable wattage reduction (up to 25%).
- Practical Approaches: Includes a simple intra-image method and an inter-image method capable of basic segmentation.


# Weakness

**Novelty Concerns**: The core inter-image approach (clustering patches into "visual words") resembles established Bag-of-Visual-Words (BoVW) and a lot of paper apply this to compress visual tokens [A,B]. The paper needs to better distinguish its novelty from these "established techniques". While the paper is comparing against ToMe, but I am not sure that is the exact difference.

> [A] "Not all patches are what you need: Expediting vision transformers via token reorganizations." arXiv preprint arXiv:2202.07800 (2022).

> [B] "Token merging: Your vit but faster." arXiv preprint arXiv:2210.09461, 2022.


**Ambiguity in Patch Representation for Clustering**: The paper applies k-means clustering to patches in "image space" but does not clearly define their representation. While it mentions that "patchification is done via basic tensor operations", this description is too vague—especially since the input to k-means is a critical methodological detail.

Given the reference to a pre-trained CLIP model (Sec. 4.1), these patches are likely derived from CLIP’s encoded features. However, this assumption should be explicitly confirmed, as the choice of representation (e.g., raw pixels, normalized embeddings, or a specific layer’s activations) fundamentally influences the clustering behavior. Key details are missing:

Which CLIP layer’s features are used? (e.g., final embeddings, intermediate projections)

Are the features normalized before clustering? (crucial for cosine distance)

Without these specifics, reproducibility and fair comparison become difficult.

**Threshold Dependency**: The inter-image approach relies on a similarity threshold (e.g., 0.1 for cosine distance ) to decide which patches to merge. While an ablation study on this threshold is provided, there is no discussion on how to determine an optimal or principled threshold for new datasets or tasks.

---

> ### Author Response · Authors · 2025-05-13
> **Rebuttal to Reviewer ZW4V**
>
> Hi, thank you for the review. We seek to address your concerns below.
>
> First, as mentioned in the related works, the VWT differs from prior approaches (e.g. [1-2]) by matching and merging similar tokens before any self-attention layers. This is a critical design choice as conducting such computations in each layer (e.g. ToME) increases the overall runtime for online inference as shown in Table 2 of our paper. Instead, we seek to compress the sequence at the beginning of the forward pass similar to text tokenizers. In this case, a significant challenge occurs as tokens only become progressively more similar to one another due to self-attention [3], which we are unable to leverage, unlike prior approaches. We overcome this via the *intra*-image (i.e. pixel variance of the individual patches) and *inter*-image (i.e. cosine distance between the bag of visual words and patches) approaches of the VWT in the pixel space.
>
> Second, as mentioned briefly above, the *intra*-image and *inter*-image approaches are executed in the pixel space. For the latter, we form the BoVW by clustering patches in the training set. During inference, similar patches are grouped by matching them to the same visual word using the cosine distance. In Table 4 of our paper, we instead show how using the embedding space (after the linear projection of tokens, but before any self-attention) is not as effective as the pixel space due to a lack of similarity between the tokens as described by [3]. As such, the VWT is novel by avoiding matching and merging in the intermediate transformer layers (i.e. more efficient) and defining two approaches to calculate token similarity without any form of fine-tuning (i.e. training-free).
>
> We will further clarify these points in the revised version of our paper.
>
> **References**
>
> [1] Not all patches are what you need: Expediting vision transformers via token reorganizations
>
> [2] Token merging: Your vit but faster
>
> [3] Patch slimming for efficient vision transformers

---

### Review · Reviewer_ujRT · 2025-05-09

**Summary Of Contributions:**

The main goal of this paper is to develop a method that can reduce the length of the input token sequence for vision transformer, in order to reduce power consumption. The key idea consists of two components. One is to mask out image patches where the pixel variance is small. The second is to first cluster image patches into a dictionary, and then use the dictionary vocabularies as replacing tokens and drop those patches that have small cosine similarities to the vocabularies. Experimental results show that these two schemes reduce the token sequence length, which results in reduction in wattage.

**Audience:**

Yes

**Broader Impact Concerns:**

I have no concern in the ethical implications of this work.

**Claims And Evidence:**

No

**Requested Changes:**

1. The authors should revise the introduction and related work to include the following information and discussion: (a). why is reducing power (it is not **energy consumption** if I understand correctly) important; (b). why this will lead to increase in processing time and why the proposed method addresses this problem.
2. The authors should explain the mechanism of pynvml GetPowerUsage function call, so we can make sure we are measuring the correct metric. How does this function calculate / estimate / measure the power. Is the power accurate? Why power matters more than energy? How does it communicate with the hardware to get the power consumption.
3. It confuse me so much why Q8 increases running time. The authors should explain.
4. The authors should also explain why the proposed method can reduce power.
5. Explain "producing shorter sequences across various distributions". What does "across various distributions" mean?

**Strengths And Weaknesses:**

**Strength**
1. The idea of masking out smooth patches and use cluster centroids to replace image patch tokens is intuitive.
2. The paper shows comprehensive and informative experimental results with adequate ablation studies, which provide insights to vision transformer researches.

**Weakness**
1. The general rationale of this research is unclear, making the research itself look less motivated. The authors seems to suggest that reducing the length of input image patch tokens will reduce power consumption but increase the inference time, which sounds strange to me. Intuitively I can see that if there is a power limit (e.g. throttle in hardware tuning), the computation time will increase. But the authors are trying to reduce the complexity. Since this is confusing, I am now not sure about the main rationale of this research.
2. Although the authors reviewed a series of related work, it is unclear to me what the gap in research is. The authors didn't distinguish this work to existing works well.
3. (Minor) The organization of the presentation of the paper makes it less readable. The tables are deliberately moved too much in the front. When readers read the paper, they need to always go back to check the tables.

---

> ### Author Response · Authors · 2025-05-14
> **Rebuttal to Reviewer ujRT**
>
> Hi, thank you for the review. We seek to address your concerns below.
>
> 1a: We refer to energy efficiency in our paper in the *general sense* of lower power consumption. By definition, power is the rate at which energy is used. Hence, a lower power consumption means less energy used over time. This is important due to the high energy cost of deploying such foundation models, particularly for large-scale inference tasks.
>
> 1b: Various approaches as listed in the related works for improving the efficiency of the vision transformer result in increased runtimes due to the additional computations required for identifying and removing redudant information within the sequence. For example, ToME [1] requires the bipartite soft matching of attention keys in every self-attention layer in order to determine the tokens to be merged with one another.
>
> 2: The function "pynvml.nvmlDeviceGetPowerUsage(handle)/1000" returns the power usage of the GPU in watts and is part of the NVIDIA Management Library, which monitors and manages NVIDIA GPU devices. We also conduct our experiments in an isolated environment, thus ensuring the measurements are as accurate as possible.
>
> 3: The 8-bit quantization introduced by [2] consists of quantize and dequantize operations to balance both performance and efficiency. The regular features have to be quantized and dequantized at every layer where matrix multiplications occur before combining with the outputs of the processed outlier features. This induces a signficant drawback to runtime despite the large gains in energy efficiency.
>
> 4: The VWT improves energy efficiency by reducing the number of tokens that have to be processed by the subsequent self-attention layers, which are quadratic with respect to the sequence length. Our token compression also occurs in the begining of the forward pass (i.e. before any self-attention layers), unlike other approaches (e.g. ToME) that do so progressively in every self-attention layer, hence improving overall runtime.
>
> 5: The sentence "producing shorter sequences across various distributions" in our paper means that the text tokenizer of newer large language models is able to effectively compress the input sequence when applied to various datasets.
>
> We will further clarify these points in the revised version of our paper.
>
> **References**
>
> [1] Token merging: Your vit but faster
>
> [2] LLM.int8(): 8-bit Matrix Multiplication for Transformers at Scale

---

### Author Response · Authors · 2025-05-20
**Revision of Paper**

Hi, we have revised our paper following the given reviews. Specifically, we have:

1. Added additional description of the wattage calculation process for further clarity.

2. Changed the term "image space" to "pixel space" for further clarity.

3. Changed the subsection title "Random Matching of the Visual Words" to "Random Merging of Tokens" for further clarity.

4. Added in the Appendix, the subsections B.1 (and Table 9) for the random dropping of tokens and B.2 (and Table 10) for the use of VWTs with other compression techniques.

5. Improved terminology used in subsection 4.6 for further clarity.

---

### Decision · Action_Editor_A1Gn · 2025-06-18

**Recommendation:** Reject

**Audience:**

Yes

**Audience Explanation:**

Yes, this topic is relevant to energy-efficient machine learning and its applications in image classification and captioning, which may interest a broad audience.

**Claims And Evidence:**

No

**Claims Explanation:**

The paper proposes a novel bottom-up approach to tokenizing images by grouping visual subwords. The central claim and contribution are not clearly stated in the introduction. According to the abstract, the method is designed for efficient online inference with minimal performance degradation. The authors evaluate their approach on benchmarks across visual recognition, robustness testing, and visual captioning, and report per-sample efficiency in terms of wattage consumption and inference runtime.

Experimental results show that the proposed method reduces token sequence length. However, Reviewer ujRT raised concerns about the energy-saving claim. Specifically, the reported 25% reduction in wattage is accompanied by up to a 20% increase in runtime, making it unclear whether overall energy consumption is indeed reduced. Table 2 appears to support a reduction in wattage, but due to the increased runtime, the AE cannot confidently verify that total energy usage is lowered.

The same reviewer also noted that the method for computing power-related metrics remains unclear even after the rebuttal, further weakening the empirical support for the energy efficiency claim.

Ultimately, while the reviewers acknowledged the importance of the problem and the interestingness of the idea, Reviewer ujRT recommends rejection. Given that this unresolved concern directly affects the paper’s core claim, the AE agrees that the manuscript is premature for acceptance in its current form. The AE also encourages the authors to revise the introduction to clearly articulate their main claims.

**Resubmission Of Major Revision:**

The authors may consider submitting a major revision at a later time.